# Direct oceanic emissions unlikely to account for the missing source of atmospheric carbonyl sulfide

Sinikka T. Lennartz[1], Christa A. Marandino[1], Marc von Hobe[2], Pau Cortes[3], Birgit Quack[1], Rafel Simo[3], Dennis Booge[1], Andrea Pozzer[4], Tobias Steinhoff[1], Damian L. Arevalo-Martinez[1], Corinna Kloss[2], Astrid Bracher[5,6], Rüdiger Röttgers[7], Elliot Atlas[8], and Kirstin Krüger[9]

[1]GEOMAR Helmholtz-Centre for Ocean Research Kiel, Düsternbrooker Weg 20, 24105 Kiel, Germany
[2]Forschungszentrum Jülich GmbH, Institute of Energy and Climate Research (IEK-7), Wilhelm-Johnen-Strasse, 52425 Jülich, Germany
[3]Institut de Ciencies del Mar, CSIC, Pg. Maritim de la Barceloneta, 37-49, 08003 Barcelona, Catalonia, Spain
[4]Max-Planck-Institute for Chemistry, Hahn-Meitner-Weg 1, 55128 Mainz, Germany
[5]Alfred-Wegener-Institute Helmholtz Center for Polar and Marine Research, Bussestrasse 24, 27570 Bremerhaven, Germany
[6]Institute of Environmental Physics, University of Bremen, 28334 Bremen, Germany
[7]Helmholtz-Zentrum Geesthacht, 21502 Geesthacht, Germany
[8]Rosenstiel School of Marine and Atmospheric Science, Miami, FL 33149 Florida, USA
[9]University of Oslo, Department of Geosciences, 0315 Oslo, Norway

*Correspondence to:* S. T. Lennartz (slennartz@geomar.de)

**Abstract.** The climate active trace-gas carbonyl sulfide (OCS) is the most abundant sulfur gas in the atmosphere. A missing source in its atmospheric budget is currently suggested, resulting from an upward revision of the vegetation sink. Tropical oceanic emissions have been proposed to close the resulting gap in the atmospheric budget. We present a bottom-up approach including (i) new observations of OCS in surface waters of the tropical Atlantic, Pacific and Indian oceans and (ii) a further improved global box model to show that direct OCS emissions are unlikely to account for the missing source. The box model suggests an undersaturation of the surface water with respect to OCS integrated over the entire tropical ocean area and, further, global annual direct emissions of OCS well below that suggested by top-down estimates. In addition, we discuss the potential of indirect emission from $CS_2$ and DMS to account for the gap in the atmospheric budget. This bottom-up estimate of oceanic emissions has implications for using OCS as a proxy for global terrestrial $CO_2$ uptake, which is currently impeded by the inadequate quantification of atmospheric OCS sources and sinks.

## 1 Introduction

Carbonyl sulfide (OCS) is the most abundant reduced sulfur compound in the atmosphere. It enters the atmosphere either by direct emissions, e.g. from oceans, wetlands, anoxic soils or anthropogenic emissions, or indirectly via oxidation of the short-lived precursor gases dimethylsulfide (DMS) and carbon disulfide ($CS_2$) (Chin and Davis, 1993; Watts, 2000; Kettle, 2002). Both precursor gases are naturally produced in the oceans, and $CS_2$ has an additional anthropogenic source (Kettle, 2002; Stefels et al., 2007; Campbell et al., 2015). Combining direct and indirect marine emissions, the ocean is considered as the dominant source of atmospheric OCS (Chin and Davis, 1993; Watts, 2000; Kettle, 2002). The most important sink

of atmospheric OCS is uptake by terrestrial vegetation (Brown and Bell, 1986; Protoschill-Krebs and Kesselmeier, 1992; Campbell et al., 2008) and oxic soils, while chemical loss by photolysis and reaction with the hydroxyl radical (OH) in the atmosphere are minor loss processes (Chin and Davis, 1993; Watts, 2000; Kettle, 2002). While tropospheric volume mixing ratios show a distinct annual cycle (Montzka et al., 2007), the interannual to decadal variation is low (Montzka et al., 2007; Kremser et al., 2015).

Accurate accounts of sources and sinks of atmospheric OCS are crucial for two reasons.

- First, OCS is climate-relevant because it influences the radiative budget of the Earth as a greenhouse gas and by contributing significant amounts of sulfur to the stratospheric aerosol layer (Crutzen, 1976; Brühl et al., 2012; Notholt et al., 2003; Turco et al., 1980) that exerts a cooling effect (Turco et al., 1980; Kremser et al., 2016). The two opposite effects are currently in balance (Brühl et al., 2012), but future changes in atmospheric circulation, as well as the magnitude and distribution of OCS sources and sinks, could change that. Hence, a better understanding of the tropospheric budget is needed to predict the effect of OCS in future climate scenarios (Kremser et al., 2016).

- Second, OCS has recently been suggested as a promising tool to constrain terrestrial $CO_2$ uptake, i.e. gross primary production (GPP), as it is taken up by plants in a similar way as $CO_2$ (Asaf et al., 2013). GPP, a major global $CO_2$ flux, can only be inferred from indirect methods, because the uptake of $CO_2$ occurs along with a concurrent release by respiration. Unlike $CO_2$, OCS is irreversibly degraded within the leaf. GPP can thus be estimated based on the uptake ratio of OCS and $CO_2$, from the leaf to regional scale (Asaf et al., 2013) or even global scale (Beer et al., 2010), under the condition that other sources are negligible or well quantified. The magnitude of terrestrial biogeochemical feedbacks on climate has been suggested to be similar to that of physical feedbacks (Arneth et al., 2010). In order to reduce existing uncertainties, it is thus crucial to better constrain single processes in the carbon cycle, especially GPP.

Nonetheless, current figures for tropospheric OCS sources and sinks carry large uncertainties (Kremser et al., 2016). While the budget has been previously considered closed (Kettle, 2002), a recent upward revision of the vegetation sink (Sandoval-Soto et al., 2005; Suntharalingam et al., 2008; Berry et al., 2013) led to a gap, i.e. a missing source in the atmospheric budget of 230-800 Gg S per year (Suntharalingam et al., 2008; Berry et al., 2013; Kuai et al., 2015; Glatthor et al., 2015) (Tab. 1), with the most recent estimates at the higher end of the range. This revision of vegetation uptake was suggested to (i) take into account the different deposition velocities of $CO_2$ and OCS within the leaf and base it on GPP instead of net primary production (Sandoval-Soto et al., 2005) as well as (ii) to better reproduce observed seasonality of OCS mixing ratios in several atmospheric models (Berry et al., 2013; Kuai et al., 2015; Glatthor et al., 2015). Based on independent top-down approaches using MIPAS (Glatthor et al., 2015) and TES (Kuai et al., 2015) satellite observations, FTIR measurements (Wang et al., 2016) as well as NOAA ground based time series stations and the HIPPO aircraft campaign (Berry et al., 2013; Kuai et al., 2015), the missing source of OCS was suggested to originate from the (tropical) ocean, most likely from the region of the Pacific warm pool. Other potential sources like e.g. advection of air masses from Asia have been discussed (Glatthor et al., 2015), but not tested. If the ocean was to account for the missing source, the total top-down oceanic source strength would then be the *a priori* oceanic flux plus the missing source estimate of each inverse model simulation (Tab. 1). This addition would imply a

200-380% increase of the *a priori* estimated oceanic source. If oceanic direct and indirect emissions were to account for the total missing source, an ocean source strength of 465-1089 Gg S yr$^{-1}$ would be required (Tab. 1).

OCS and its atmospheric precursors are naturally produced in the ocean. In the surface open ocean, OCS is present in the lower picomolar range, and has been measured on numerous cruises in the Atlantic (Ulshöfer et al., 1995; Flöck and Andreae, 1996; Ulshöfer and Andreae, 1998; von Hobe et al., 1999), including 3 latitudinal transects (Kettle et al., 2001; Xu et al., 2001), the Indian Ocean (Mihalopoulos et al., 1992), the Pacific Ocean (Weiss et al., 1995a) and the Southern Ocean (Staubes and Georgii, 1993). Measurements in tropical latitudes, where the missing source is assumed to be located, have previously been performed in the Indian Ocean (Mihalopoulos et al., 1992) and during the Atlantic transects (Kettle et al., 2001; Xu et al., 2001). OCS is produced photochemically from chromophoric dissolved organic matter (CDOM) (Andreae and Ferek, 2002; Ferek and Andreae, 1984) and by a not fully understood light independent production that has been suggested to be linked to radical formation (Flöck et al., 1997; Pos et al., 1998). Dissolved OCS is efficiently hydrolyzed to $CO_2$ and $H_2S$ at a rate depending on pH and temperature (Elliott et al., 1989). $CS_2$ has been measured in the Pacific and Atlantic oceans in a range of 7.2-27.5 pmol L$^{-1}$ (Xie et al., 1998) and during two Atlantic transects (summer and winter) in a range of 4-40 pmol L$^{-1}$ (Xu, 2001). It is produced photochemically (Xie et al., 1998) and biologically (Xie et al., 1999), and no significant loss process other than air-sea gas exchange has been identified (Xie et al., 1998). DMS is present in the lower nanomolar range in the surface ocean and has been extensively studied in several campaigns, summarized in a climatology by Lana et al. (2011). DMS is biogenically produced and consumed in the surface ocean, as well as photo-oxidized and ventilated by air-sea exchange (Stefels et al., 2007).

Available bottom-up estimates of the global oceanic OCS fluxes from shipboard observations range from -16 Gg S yr$^{-1}$ to 320 Gg S yr$^{-1}$ (Tab. 2). However, the highest estimates were biased, because mainly summertime and daytime observations of water concentrations were considered. With the discovery of the seasonal oceanic sink of OCS during wintertime (Ulshöfer et al., 1995) and a pronounced diel cycle (Ferek and Andreae, 1984), direct oceanic emissions were corrected downwards.

Only recently, OCS emissions have been estimated with the biogeochemical ocean model NEMO-PISCES (Launois et al., 2015a) at a magnitude of 813 Gg S yr$^{-1}$, sufficient to account for the missing source. This oceanic emission inventory had been used to constrain GPP based on OCS on a global scale (Launois et al., 2015b). However, the oceanic OCS photoproduction in the ocean model included a parameterization for OCS photoproduction derived from an experiment in the North Sea (Uher and Andreae, 1997b), which might not be representative for the global ocean, as indicated by an order of magnitude lower photoproduction constants in the Atlantic ocean compared to the German Bight (Uher and Andreae, 1997a).

Here, we present new observations in all three tropical ocean basins, two of them measured with unprecedented precision and time resolution. Direct fluxes were inferred from continuous OCS measurements in the tropical Pacific and Indian Oceans, covering a range of regimes with respect to CDOM content, ultraviolet radiation (UV) and sea surface temperature (SST). These observations are used to further constrain and validate a biogeochemical box model which had previously been shown to reproduce OCS concentration in the Atlantic Ocean reasonably well (von Hobe et al., 2001). The box model is now updated from its previous global application (Kettle, 2002) by adding and further developing the most recent process parametrizations to estimate the global source strength of direct OCS emissions. The emission estimate is further complemented by discussing

the potential of indirect OCS emissions, i.e. the emissions of short-lived precursor gases $CS_2$ and DMS, to account for the gap in the budget.

## 2 Methods

### 2.1 Study sites

Several cruises were conducted to measure the trace gases OCS (OASIS, TransPEGASO, ASTRA-OMZ) and $CS_2$ (TransPEGASO, ASTRA-OMZ). Cruise tracks are depicted in Fig 1. The OASIS cruise onboard RV SONNE I to the Indian Ocean started from Port Louis, Mauritius to Male, Maledives in July and August 2014, where mainly oligotrophic waters were encountered. TransPEGASO was an Atlantic transect starting in Gibraltar leading to Buenos Aires, Argentinia and Punto Arenas, Chile. It took place in October and November 2014 and covered a variety of biogeochemical regimes. ASTRA-OMZ onboard

RV SONNE II started in Guayaquil, Ecuador and ended in Antofogasta, Chile, in October 2015. Although 2015 was an El Nino year, upwelling together with high biological production was still encountered during the cruise (Stramma et al., 2016).

### 2.2 Measurement set-up for trace gases

OCS was measured during two cruises on board the RV SONNE I (OASIS) and SONNE II (ASTRA-OMZ) with a continuous underway system similar to the one described in Arévalo-Martínez et al. (2013), at a measurement frequency of 1 Hz.

The system consisted of a Weiss-type equilibrator, through which seawater is pumped from approximately 5 m below the surface with a flow of 3-4 L min$^{-1}$. The air from the equilibrator headspace was Nafion dried and continuously pumped into an OCS-analyzer (Model DL-T-100, Los Gatos Research) that uses off axis - integrated cavity output spectroycopy (OA-ICOS) technique. The instrument used on board is a prototype of a commercial instrument (www.lgrinc.com/documents/OCS_ Analyzer_ Datasheet.pdf), developed by Los Gatos Research (LGR) in collaboration with Forschungszentrum Jülich GmbH

(Schrade, 2011). Data were averaged over 2 minutes, achieving a precision of 15 ppt. OCS mixing ratios in the MBL were determined by pumping outside air ca. 50 m from the ship's deck to the OCS analyzer (KNF Neuberger pump). A measurement cycle consisted of 50 min water sampling and 10 min air sampling, where the first 3 minutes after switching until stabilization of the signal were discarded.

Before and after the cruise the analyzer was calibrated over a range of concentrations using permeation devices. Both

calibrations were consistent. However, during calibration the output of the internal spectral retrieval differed significantly from post processing of the recorded spectra, which matched the known concentrations (this offset is not present in the commercial instruments). The calibration data were thus used to derive a correction function. After correction all data stayed within 5% of the standards. The calibration scale of the permeation devices was 5% below the NOAA scale. As the OCS analyzer measured $CO_2$ simultaneously, and $CO_2$ standards were available during the cruise, drift of the instrument was tested by measuring

$CO_2$ standard gases before and after the cruise and found to be less than 1% of the signal. Special care was taken to avoid contamination and all materials used have been tested for contamination before use.

During OASIS, the mirrors inside the cavity of the OCS analyzer were not completely clean, which led to a reduced signal. To correct the data, an attenuation factor was determined from simultaneous $CO_2$ measurements, because no OCS standard was available onboard, and OASIS data was corrected accordingly.

An independent quality check of the data was performed by comparing volume mixing ratios of the MBL from the OCS analyzer with samples from air canisters sampled during both cruises and measured independently (Schauffler et al., 1998; de Gouw et al., 2009). The calibrated (and attenuation corrected for OASIS) OA-ICOS data were on average 5% lower than the air canister samples, which reflects the 5% difference between the calibration at Forschungszentrum Jülich and the NOAA scale.

During ASTRA-OMZ, $CS_2$ was directly measured on board within 1 hour of collection using a purge and trap system attached to a gas chromatograph and mass spectrometer (GC/MS; Agilent 7890A/Agilent 5975C; inert XL MSD with triple axis detector) running in single ion mode. The discrete surface seawater samples (50 mL) were taken each hour to every three hours from the same pump system as for continuous OCS measurements. $CS_2$ was stripped by purging with helium (70 mL min$^{-1}$) for 15 minutes. The gas stream was dried using a Nafion membrane dryer (Perma Pure) and $CS_2$ was preconcentrated in a trap cooled with liquid nitrogen. After heating the trap with hot water, $CS_2$ was injected into the GC/MS. Retention time for $CS_2$ (m/z 76, 78) was 4.9 minutes. The analyzed data was calibrated each day using gravimetrically prepared liquid $CS_2$ standards in ethylene glycol. While purging, 500 $\mu$L gaseous deuterated DMS (d3-DMS) and isoprene (d5-isoprene) were added to each sample as an internal standard to account for possible sensitivity drift between calibrations.

During the TransPEGASO cruise on board R/V Hesperides, surface ocean OCS and $CS_2$ were measured in discrete seawater samples by purge and trap and gas chromatography with mass spectrometry detection (GC-MSD). Samples were collected every day at 9:00 and 15:00 h local time in glass bottles without headspace and analyzed within 1 hour. Aliquots of 25 mL were withdrawn with a glass syringe and filtered through GF/F while injected into the purge and trap system (Stratum, Teledyne Tekmar). The water was heated to 30°C and volatiles were stripped by bubbling with 40 mL min-1 of ultrapure helium for 12 minutes and trapped in a U-shaped VOCARB 9 trap at room temperature. After flash thermal desorption, volatiles were injected into an Agilent 5975T LTM GC-MSD equipped with an Agilent LTM DB-VRX column (20 m x 0.18 mm OD x 1$\mu$m) maintained at 30°C. Retention times for OCS (m/z 60) and $CS_2$ (m/z 76) were 1.3 and 2.7 min, respectively. Peak quantification was achieved with respect to gaseous (OCS in $N_2$) and liquid ($CS_2$ in methanol and water) standards that were analyzed in the same way. Samples were run in duplicates. Detection limits were 1.8 pM (OCS) and 1.4 pM ($CS_2$), and precision was typically around 5%.

The systems are calibrated against a standard each, but had not been directly intercompared. Still, our measurements are consistent with previous measurements using independent methods as discussed in section 3.2.1 and 3.3.

## 2.3  Calculation of air-sea exchange

Fluxes F of all gases were calculated with Eq. 1:

$$F = k_w \cdot \Delta C \tag{1}$$

where $k_w$ is the gas transfer velocity in water (i.e. physical constraints on exchange) and $\Delta C$ the air-sea concentration gradient (i.e. the chemical constraint on exchange). The air-side transfer velocity (Liss and Slater, 1974) for OCS was calculated to be seven orders of magnitude smaller and was therefore neglected. The concentration gradient was determined using the temperature dependent Henry constant (De Bruyn et al., 1995) and the measurements in the surface water and marine boundary layer (MBL) for OASIS and ASTRA-OMZ. During TransPEGASO, no atmospheric volume mixing ratio was measured, and a value of 500 ppt was assumed (Montzka et al., 2007). As air volume mixing ratios of OCS vary over the course of a year, we performed a sensitivity test for a scenrio of 450 and 550 ppt and found mean deviations of +7.8 and -7.8 % respectively. The transfer velocity $k_w$ was determined using a quadratic parameterization based on wind speed (Nightingale et al., 2000) which was directly measured onboard (10 minute averages). Furthermore $k_w$ was corrected for OCS and $CS_2$ by scaling it with the Schmidt number calculated from the molar volume of the gases (Hayduk and Laudie, 1974). It should be noted that the choice of the parameterization for $k_w$ has a non-negligible influence on the global emission estimate. Linear, quadratic and cubic parameterizations of $k_w$ are available, with differences increasing at high wind speeds in the order of a factor of 2 (Lennartz et al., 2015; Wanninkhof et al., 2009). Evidence suggests that the air-sea exchange of insoluble gases such as $CO_2$, OCS and $CS_2$, follows a cubic relationship to wind speed because of bubble-mediated gas transfer (McGillis et al., 2001; Asher and Wanninkhof, 1998). However, this difference between soluble and non-soluble gases is not always consistent (Miller et al., 2009), and too little data is available for a reliable parameterization at high wind speeds above $12\,\mathrm{m\,s^{-1}}$, where the cubic and the quadratic parameterizations diverge the most. For reasons of consistency, e.g. for the fitted photoproduction $p$ from previous studies, and the fact that most of the previous emission estimates were computed using a quadratic $k_w$ parameterization, we chose the same quadratic parameterization representing the mean range of observations (Nightingale et al., 2000). For a sensitivity test, we computed the global oceanic emission with a cubic relationship (McGillis et al., 2001), which results in an additional 40 Gg S per year as direct OCS emissions, leaving the missing source still unexplained. However, better constraints on the transfer velocity of insoluble gases would decrease the uncertainty of global oceanic emissions of marine trace gases.

## 2.4 Box model of OCS concentration in the surface ocean

A box model to simulate surface concentration of OCS is further developed from the latest version from von Hobe et al. (2003, termed vH2003), where concentrations along the cruistracks of 5 Atlantic cruises have been simulated and compared. The vH2003 model results from successful tests and validation to observations on several cruises to the Altantic Ocean covering all seasons (i.e. Flöck and Andreae (1996) in January 1994, Uher and Andreae (1997a) in April/May 1992, von Hobe et al. (1999) in June/July 1997, Kettle et al. (2001) in September/October 1998). By comparing photoproduction rate constants of the 5 cruises to CDOM absorption, von Hobe et al. (2003) suggests a second order process for photoproduction with the photoproduction rate constant being dependent on the absorption of CDOM in seawater.

In our approach, we test vH2003 along the cruise track of two cruises, include a new way of determining the photoproduction rate constant (see below) and apply it with global climatological input (termed L2016). Kettle (2000, 2002, termed K2000) applied a similar version of vH2003 globally, which included an optimized photoproduction constant from Atlantic transect cruise data, an optimized constant light-independant production and a linear regression to obtain CDOM from chlorophyll $a$.

In comparison to K2000, we use (i) a new way of determining the photoproduction rate constant incorporating information from three ocean basins, (ii) the most recent parameterization of light-independent production available, and (iii) satellite observations for sea surface CDOM instead of an empirical relationship based on chlorophyll *a*.

Launois et al. (2015a) implemented parameterizations for light-independant production, hydrolysis and air-sea exchange similar to vH2003 in the 3D global ocean model NEMO-PISCES. The main differences to the approach used here is the lack of accounting for mixing in L2016 (discussed in section 3.2.2, which will theoretically lead to higher simulated concentrations in our case) and the application of a photoproduction rate constant in our model that incorporates information from three open ocean basins in contrast to one from a study in the North Sea (Launois et al., 2015a).

In L2016, the light-independent production term of OCS was parameterized depending on SST [K] and the absorption coefficient of CDOM at 350nm wavelength, $a_{350}$ (von Hobe et al., 2001) (Eq. 2).

$$\frac{dC_{OCS}}{dt} = a_{350} \cdot 10^{-6} \cdot exp(55.8 - \frac{16200}{SST}) \tag{2}$$

The parameterization for hydrolysis describes alkaline and acidic degradation of OCS by the reactions R1 and R2:

$$OCS + H_2O \rightarrow H_2S + CO_2 \tag{R1}$$

$$OCS + OH^- \rightarrow SH^- + CO_2 \tag{R2}$$

It was parameterized as a first order kinetic reaction including the rate constant $k_h$ according to Eq. 3-5:

$$\frac{dC_{OCS}}{dt} = [OCS] \cdot k_h \tag{3}$$

$$k_h = exp(24.3 - \frac{10450}{SST}) + exp(22.8 - \frac{6040}{SST}) \cdot \frac{K}{a[H^+]} \tag{4}$$

$$-log_{10}K = \frac{3046.7}{SST} + 3.7685 + 0.0035486 \cdot \sqrt{SSS} \tag{5}$$

where $a[H^+]$ is the proton activity and $K$ the ion product of seawater (Dickinson and Riley, 1979).

Fluxes were calculated with Eq. 1 using the same parameterization for $k_w$ as for the emission calculation from measurements described above.

Photoproduction was integrated over the mixed layer depth (MLD), assuming a constant concentration of OCS and CDOM throughout the mixed layer, with the photoproduction rate constant $p$ [mol J$^{-1}$], $a_{350}$ [m$^{-1}$] and the UV [W m$^{-2}$] (Sikorski and Zika, 1993) (Eq. 6).

$$\frac{dC_{OCS}}{dt} = \int_{-MLD}^{0} pa_{350}UV\,dz \tag{6}$$

MLD was obtained from CTD (conductivity, temperature, depth) profiles and interpolated between these locations (S-Fig 1,2). The photochemically active radiation that reaches the ocean surface was approximated by Eq. 7 (Najjar et al., 1995):

$$UV = 2.85 \cdot 10^{-1} \cdot I \cdot cos^2\theta \tag{7}$$

with global radiation $I$ [W m$^{-2}$] and the zenith angle cos $\theta$. The attenuated UV light intensity directly below the surface

(Sikorski and Zika, 1993) down to the respective depth of the mixed layer was calculated in 1 m steps, taking into account attenuation by CDOM and pure seawater. As a simplification in this global approach, the box model did not resolve the whole wavelength spectrum, but rather used a$_{350}$ and applied a photoproduction rate constant that takes into account the integrated spectrum. A similar approach had been tested and compared to a wavelength spectrum resolving version by von Hobe et al. (2003).

The rate coefficients for hydrolysis, light-independent production and air-sea exchange are all reasonably well constrained and parameterizations have been derived from dedicated laboratory and field experiments (hydrolysis, air-sea exchange) or from nighttime OCS observations in several regions assuming steady-state (dark production, von Hobe et al. (2001)). On the contrary, the photoproduction rate constant $p$ is not well constrained and no generally applicable parameterization exists. von Hobe et al. (2003) have made a start of parameterizing $p$ in terms of CDOM absorption, and found this to be dependent on the

exact model setup used with respect to wavelength integration and mixed layer treatment. To extend the $p$-CDOM-relationship for other ocean basins, we use the two cruises OASIS and ASTRA-OMZ as case studies for parameter optimization of the photoproduction rate constant $p$. The photoproduction constant $p$ in the case study simulations was fitted individually for periods of daylight >100 W m$^{-2}$ (Fig. 2, blue lines) with a Levenberg-Marquart optimization routine in MatLab version 2015a (8.5.0), by minimizing residuals between simulated and hourly averaged measurements. Different starting values were tested

to reduce the risk of the fitted $p$ being a local minimum. Together with photoproduction rate constants obtained by a similar optimization procedure by von Hobe et al. (2003) (Tab. 2 therein, termed MLB STC), a relationship of the photoproduction constant $p$ dependent on a$_{350}$ was established (Fig. 3). The resulting linear relationship thus includes values from the Altantic, Pacific and Indian Ocean, making it a good approximation for a globally valid dependence. For the global box model, $p$ was calculated in every time step based on this relationship (r=0.71,Eq. 8):

$$p = 3591.3 \cdot a_{350} + 329.4 \tag{8}$$

The scatter in Fig. 3 likely reflects the inhomogeneity of the water masses across the three oceanic basins considered, as CDOM absorbance is a valid proxy, but carries some uncertainty in the concentration of the actual precursor.

The model input for simulations of the cruises OASIS and ASTRA-OMZ consisted of measurements made during the respective cruise, including SST and SSS (MicroCAT SBE41) measured every minute, CDOM absorption coefficient (spectropho-

tometrically measured ca. every 3 hours with a liquid capillary cell setup) and the ship's in situ measured meteorological data such as wind speed and global radiation averaged over 10 minutes (S-Fig. 1,2, S-Tab. 1,2). Forcing data was linearly interpolated to the time step of integration of 2 minutes.

For the global box model, monthly global meteorological fields with a spatial resolution of 2.8 x 2.8° were used (S-Tab. 3, S-Fig. 3). For global a$_{350}$ at the sea surface, monthly climatological means for absorption due to gelbstoff and detritus a$_{443}$

from the MODIS-Aqua satellite (all available data, 2002-2014) (NASA, 2014) were corrected to 350 nm with Eq. 9 (Fichot and Miller, 2010; Launois et al., 2015a):

$$a_{350} = a_{443} \cdot exp(-0.02 \cdot (350 - 443)) \tag{9}$$

SST, wind speed, and atmospheric pressure were obtained as monthly climatological means from the same period, i.e. 2002 to 2014, by ERAInterim (Dee et al., 2011). A diel cycle of global radiation $I$ was obtained by fitting the parable parameters $a$ and $b$ during time of the day $t$ in Eq. 10 (S-Fig. 4):

$$I = -a \cdot t^2 + b \tag{10}$$

to conditions of (i) x-axis interceptions in the distance of the sunshine duration and (ii) the integral being the daily incoming energy by ERAInterim (Dee et al., 2011). Monthly climatologies of mixed layer depths were used from the MIMOC project (Schmidtko et al., 2013). For details of data sources please refer to S-Tab. 1-3 provided in the supplementary material. The time step of the model was set to 120 minutes, which had been tested to result in negligible (<3%) smoothing.

## 2.5 Assessing the indirect contribution of DMS with EMAC

Model output from the ECHAM/MESSy Atmospheric Chemistry (EMAC) from the simulation RC1SDbase-10a of the ES-CiMo project (Jöckel et al., 2015) are used to evaluate the contribution of DMS on the production of OCS. The model results were obtained with ECHAM5 version 5.3.02 and MESSy version 2.51, with a T42L90MA resolution (corresponding to a quadratic Gaussian grid of approx. 2.8 by 2.8 ° in latitude and longitude) and 90 vertical hybrid pressure levels up to 0.01 hPa. The dynamics of the general circulation model were nudged by Newtonian relaxation towards ERA-Interim reanalysis data. DMS emissions were calculated with the AIRSEA submodel (Pozzer et al., 2006), which takes in account concentration of DMS in the atmosphere and in the ocean, following a two-layer conceptual model to calculate emissions (Liss and Slater, 1974). While atmospheric concentrations are estimated online by the model (with DMS oxidation), the oceanic concentrations are prescribed as monthly climatologies (Lana et al., 2011). It was shown that such an online calculation of emissions provides the most realistic results when compared to measurements compared to a fixed emission rate (Lennartz et al., 2015). The on-line calculated concentration of DMS and OH have then been used to estimate the production of OCS. A production yield of 0.7% has been used for the reaction of DMS with OH (Barnes et al., 1994), using the reaction rate constant suggested by the International Union of Pure and Applied Chemistry (IUPAC) (Atkinson et al., 2004).

## 3 Results and Discussion

### 3.1 Observations of OCS in the tropical ocean

OCS was measured in the surface ocean and MBL during three cruises in the tropics. Measurement locations (Fig. 1) include oligotrophic open ocean regions in the Indian Ocean (OASIS, 07-08/2014), open ocean and shelf areas in the eastern Pacific

(ASTRA-OMZ, 10/2015) and a meridional transect in the Atlantic (TransPEGASO, 10-11/2014). In the Indian and Pacific Oceans, continuous underway measurements provided the necessary temporal resolution to observe diel cycles of OCS concentrations in surface water. Dissolved OCS concentrations exhibited diel cycles with maxima 2 to 4 hours after local noon (Fig. 1), which are a consequence of photochemical production and removal by hydrolysis (Uher and Andreae, 1997a). OCS

concentrations also varied spatially. Taking $a_{350}$ as a proxy for CDOM content, daily mean OCS concentrations were higher in CDOM rich (Tab. 3, 28.3±19.7 pmol OCS L$^{-1}$, $a_{350}$: 0.15±0.03 m$^{-1}$) than in CDOM poor waters (Tab. 3, OASIS: 9.1±3.5 pmol OCS L$^{-1}$, $a_{350}$: 0.03±0.02 m$^{-1}$). Samples during TransPEGASO were measured with gas chromatography/mass spectrometry twice a day (around 8-10 and 15-17 h local times). Therefore, the full diel cycles could not be reconstructed and potential variations of OCS with CDOM absorption were overlaid by diel variations. Nevertheless, the observed range of OCS

concentrations in the Atlantic corresponds well to the observations from the eastern Pacific and Indian Ocean (Tab 3), and is consistent with measurements from a previous Atlantic meridional transect (AMT-7) cruise (Kettle et al., 2001) (1.3-112.0 pmol OCS L$^{-1}$, mean 21.7 pmol OCS L$^{-1}$).

Air-sea fluxes calculated from surface concentrations and mixing ratios of OCS as a function of wind speed generally follow the diel cycle of the surface ocean concentration. While supersaturation prevailed during the day, low nighttime concentrations

usually led to oceanic uptake of atmospheric OCS. OCS fluxes integrated over one day ranged from -0.024 to -0.0002 g S km$^{-2}$ in the open Indian Ocean and from 0.38 to 2.7 g S km$^{-2}$ in the coastal Pacific. During the observed periods, the ocean was a net sink of atmospheric OCS in the Indian Ocean, whereas it was a net source in the eastern Pacific. Although an assessment of net flux is difficult given the lower temporal resolution during TransPEGASO, calculated emissions were in the same range as the ones measured in the Pacific and Indian Ocean.

The water masses encountered during the cruises to the Indian Ocean (OASIS) and eastern Pacific (ASTRA-OMZ), which are used to constrain the global box model, differ considerably with respect to the properties relevant for OCS cycling and, thus, span a large range of possible OCS variability. The properties encountered during these two cruises encompass or exceed the ones of the Pacific warm pool (climatological averages, Tab. 4), which is where the location of the missing source has been hypothesized (Glatthor et al., 2015; Kuai et al., 2015). Both higher SST and lower wind speeds (Tab. 4) would decrease the

OCS sea surface concentrations in the ocean, leading to decreased emissions to the atmosphere: higher SSTs favor a stronger degradation by hydrolysis (Elliott et al., 1989), and lower wind speeds decrease the transfer velocity *k*. Lower integrated daily radiation (SR in Tab. 4) in the Pacific warm pool also points to lower OCS production. Hence, our new OCS observations presented here likely span the range of emission variability in the tropics.

The observed concentrations and calculated emissions are approximately one order of magnitude lower than the annual mean

surface concentrations and emissions simulated in the 3D global ocean model NEMO-PISCES (Launois et al., 2015a).

## 3.2   A direct global oceanic emission estimate for OCS

The OCS observations from the Indian and Pacific Ocean were used to improve a box model for simulating OCS concentrations in the surface ocean (Kettle, 2002; Uher and Andreae, 1997b; von Hobe et al., 2003). With the $a_{350}$ dependent photoproduction constant included, the model reproduced the diel pattern of OCS concentrations in the surface oceans for both cruises (Fig.

2, black lines). A slight overestimation of observed concentrations is present for the Indian Ocean cruise OASIS (observed mean concentration: 9.1±3.5 pmol L$^{-1}$; simulated: 10.8±3.9 pmol L$^{-1}$). This overestimation was more pronounced in the eastern Pacific (observed mean: 28.3±19.7 pmol L$^{-1}$; simulated: 47.3±25.4 pmol L$^{-1}$) and can largely be attributed to a lack of downward mixing inherent in the mixed layer box model due to the assumption of the OCS concentration being constant throughout the entire mixed layer.

Using the linear $p$-a$_{350}$ parameterization for the first time in a global model, the same box model as for the case studies is applied to estimate sea surface concentrations and fluxes of OCS on a global scale (Fig. 4). The OCS production is consistent with the global distribution of CDOM absorption (S-Fig. 5) with highest concentrations calculated for coastal regions and higher latitudes. Despite the photochemical hotspot in the tropics (30°N-30°S), degradation by hydrolysis prevents any accumulation of OCS in the surface water, as we calculated the lifetime due to hydrolysis to be only 7 hours (S-Fig. 5). The simulated range of water concentrations is too low to sustain emissions in the tropics that could close the atmospheric budget of OCS (Fig. 4). Integrated over one year, the tropical ocean (30°N-30°S) is even undersaturated with respect to OCS, uptaking 3.0 Gg S yr$^{-1}$. Globally, the integration over one year yields annual oceanic OCS emissions of 130 Gg S. Our results corroborate the upper limit of an earlier study that used an observation-derived emission inventory (Tab. 1) (Kettle, 2002), but includes more process oriented parametrizations as described in section 2.4. Clearly, our results from both observations and modelling contradict the latest bottom-up emission estimate from the NEMO-PISCES model (Launois et al., 2015a), and do not support a hot spot of direct OCS emissions in the Pacific Warm Pool or the tropical oceans in general.

### 3.2.1 Comparison to previous shipbased measurements

The global simulation of OCS surface water concentrations generally reproduced the lower picomolar range of concentrations (Tab. 5), the seasonal pattern of higher concentrations during summer compared to winter (as e.g. in Ulshöfer et al., 1995) and the spatial pattern of higher concentrations in higher latitudes (e.g. Southern Ocean, Staubes and Georgii, 1993). Given that monthly means of a model simulation driven by climatological data of the input parameters is compared to cruise measurements, the absolute mean deviation of 6.9 pmol L$^{-1}$ and the mean deviation of 3.7 pmol L$^{-1}$ indicate an overall good reproduction of observations (differences between observation and model output were weighted to number of observations in Tab. 4). It should be noted that, on average, the model overestimates OCS concentrations as indicated by the positive mean error, suggesting our emission estimate to be an upper limit to direct oceanic OCS emissions in most regions. The largest deviation from observations are found in the Southern Ocean (vgl. Staubes and Georgii, 1993 in Tab. 4), where the model underestimared observations by 40%. While this can be due to several explanations, i.e. a possible violation of the underlying assumption of a constant OCS production in regions with deep mixed layers such as the Southern Ocean, or the missing satellite data for CDOM during polar nights, it is a clear indication of the need of more observations from high latitudes. However, this underestimation does not interfere with our conclusion drawn for the tropical oceans, where the location of the missing source is derived from top-down approaches.

### 3.2.2 Uncertainties

Simulated concentrations and fluxes carry uncertainties from input parameters and process parameterizations. One major uncertainty associated with the mixed-layer box model approach arises from the fact that it does not adequately account for downward mixing and vertical concentration gradients within the mixed layer. Under most circumstances, and especially in the tropical open ocean, where hydrolysis greatly exceeds surface outgassing and low $a_{350}$ makes photoproduction extend further down in the water column, the model tends to overestimate the real OCS concentrations, as was shown for our two cruises above. Therefore, we deem the fluxes from our global simulation to represent an upper limit of the true fluxes. Only at high latitudes would we expect more complex uncertainties, because hydrolysis at low temperatures is slow and only photoproduction and loss by outgassing are directly competing at the very surface.

Other uncertainties are associated with the calculation of the photoproduction rate. The wavelength of 443nm combines the absorption of detritus and CDOM, which could have an impact especially in river plumes, where terrestrial material is transported into the ocean. As it is the CDOM that is important for photochemistry, assuming the 443nm is purely CDOM would lead to an overestimation of photoproduction and therefore is a conservative estimate. It should also be noted that a single spectral slope from 443nm to 350nm in the global simulation is a simplification. Furthermore, using a wavelength integrated photoproduction rate constant instead of a wavelength resolved approach, which would take global variations in the CDOM and light spectra into account, is an additional simplification. It has been shown that this does not lead to large differences regionally (von Hobe et al., 2003), but, potentially, could lead to variations globally. Our p-CDOM-relationship is a first step for constraining this variability globally in one parameterization, as it incorporates photoproduction rate constants optimized to observations and thus accounting for differences in the light and CDOM spectra. More data from different regions can help to further constrain this relationship in future studies. Despite these simplifications, the simulated concentrations agree very well with previous observations (n>4000, Tab. 4). To test the sensitivity of our box model to the photoproduction rate constant, we performed a sensitivity test with a photoproduction increased by a factor of 5 in the tropical region (30°N-30°S, note that this factor is considerably larger than the uncertainty in the p-CDOM-relationship). This leads to an annual mean concentration of 35.1 pmol L$^{-1}$ in the tropics (30°N-30°S), resulting in tropical direct emissions of 160 Gg S as OCS per year. The efficient hydrolysis in warm tropical waters prevents OCS concentrations from accumulating despite the high photoproduction, and still results in emissions too low to account for the missing source.

With a mean error of 3.7 pmol L$^{-1}$ in the OCS surface water concentrations added to (subtracted from) the modelled concentration and subsequent calculation of fluxes using annual climatologies for wind, pressure and SST (same data sources as global simulation forcing data), we calculate an uncertainty of 60%, which translates into a total uncertainty of the integrated global flux of 80 Gg S yr$^{-1}$.

### 3.3 Indirect OCS emissions by DMS and CS$_2$

A significant contribution to the OCS budget in the atmosphere results from oceanic emissions of DMS and CS$_2$ that are partially converted to OCS on time scales of hours to days (Chin and Davis, 1993; Watts, 2000; Kettle, 2002). A yield of 0.7 %

for OCS is used for the reaction of DMS with OH (Barnes et al., 1994), which results in a global oceanic source of DMS from OCS of 80 (65 - 110) Gg S yr$^{-1}$ based on the procedure discribed in section 2.5. The uncertainty range of 65-110 Gg S yr$^{-1}$ originated from the uncertainty in oceanic emissions, not the conversion factor. This conversion factor is much more uncertain, as the formation of OCS from DMS involves a complex multi-step reaction mechanism that is not fully understood. It has been

shown in laboratory experiments, that the presence of NO$_x$ reduces the OCS yield considerably (Arsene et al., 2001), which would make our indirect emission estimate an upper limit. However, the yield was measured under laboratory conditions, and may be different and more variable under natural conditions.

DMS emissions do not show a pronounced hot spot in the Pacific warm pool region, but as DMS transports much more sulfur across the air-sea interface than OCS, even low changes in the OCS yield could affect the atmospheric budget of OCS. As the

10 spatial oceanic emission pattern of DMS does not reflect the spatial pattern of the assumed missing source, a locally specific tropospheric change in the conversion yield would be one potential way of bringing the patterns in agreement. While it is possible that the OCS yield could vary under certain conditions, e.g. it cannot be excluded that the low OH concentrations in the broader Pacific warm pool area as suggested by Rex et al. (2014) influence the yield, the (local) increase of the conversion factor would need to be in the order of a factor of 10-100.

For CS$_2$, the atmospheric reaction pathway producing OCS is better understood with a well constrained molar conversion ratio of 0.81 (Chin and Davis, 1993). However, the global distribution of oceanic CS$_2$ concentration, hence its emissions to the atmosphere, are poorly known. In our study, surface CS$_2$ concentrations (S-Fig. 6) were on average 17.8±8.9 pmol L$^{-1}$ during ASTRA-OMZ, and 62.5±42.1 pmol L$^{-1}$ during TransPEGASO (Tab. 3). The latter values are higher than previously reported concentrations from the AMT-7 cruise in the central Atlantic (Kettle et al., 2001) (10.9±15.2 pmol L$^{-1}$). We extrapolate a

weighted mean of the CS$_2$ emissions from TransPEGASO (n=42, 13.7±9.8 g S d$^{-1}$ km$^{-2}$), ASTRA-OMZ (n=122, 4.1±3.2 g S d$^{-1}$ km-2) and AMT-7 (Kettle et al., 2001) (n=744, 1.6±1.8 g S d$^{-1}$ km$^{-2}$) in order to estimate CS$_2$ derived OCS emissions from the global ocean. According to our extrapolation, 135 (7-260) Gg S yr$^{-1}$ enter the atmosphere as oceanic CS$_2$ emissions converted to OCS. The uncertainty range of 7-260 Gg S yr$^{-1}$ results from extrapolating the highest and the lowest emissions encountered during the cruises to the global ocean. This number is at the highest end of the range for CS$_{2OCS}$ emissions

from globally simulated CS$_2$ oceanic concentrations (Kettle, 2000, 2002), as measured CS$_2$ concentrations from the cruises ASTRA-OMZ and TransPEGASO are higher than the simulated surface concentrations in Kettle (2000) for the respective month. However, the spatial pattern of higher concentrations and emissions in the tropical region in our measurements agrees well with the spatial pattern simulated in Kettle (2000). Nonetheless, even the extrapolation of the highest measurement would not close the budget for the three largest missing source estimates (Tab. 1).

For oceanic emission estimates used to constrain GPP, quantifying the seasonal cycle of the single contributors is essential. For example, high emissions during oceanic spring and fall blooms could mask OCS uptake by the terrestrial vegetation, and therefore neglecting them could lead to an underestimation of global GPP, with implications for the atmospheric and terrestrial carbon budget.

## 4 Conclusions and Outlook

Considering the observational evidence and the modelled global emission estimate of $130\pm80$ Gg S yr$^{-1}$, direct OCS emissions from the oceans are too low to account for the missing atmospheric source. Together with indirect emissions, the oceanic source strength of OCS would add up to 345 Gg S yr$^{-1}$ compared to 465-1089 Gg S yr$^{-1}$ required to balance the suggested increase of vegetation uptake. Direct and even additional indirect oceanic emissions of OCS are thus unlikely to balance the budget after the upward revision of the vegetation sink. Largest uncertainties are associated with the indirect emission estimates, especially in the conversion of DMS to OCS and the global source strength of $CS_2$.

As our study suggests, the search for an additional source of OCS to the atmosphere should include other sources than oceanic emissions alone. There are indications of other parts of the OCS budget being underestimated, such as domestic coal combustion (Du et al., 2016). While biomass burning is known to emit OCS and is present close to the assumed source region, e.g. around Indonesia, the most recent review of OCS emission factors result in a source too small to close the atmospheric budget (Campbell et al., 2015). However, Lee and Brimblecombe (2016) reevaluated the anthropogenic emissions of OCS and its precursors and provide a higher number than previously considered of 598 Gg S yr$^{-1}$. They attribute the largest direct OCS emissions to biomass and biofuel burning, as well as pulp and paper manufactoring, and the largest $CS_2$ emissions to the rayon industry. Hence, a hot spot of anthropogenic emissions in the Asian continent might be a potential candidate, together with atmospheric transport, to produce atmospheric mixing ratios as observed by satellite.

A redistribution of the magnitude and seasonality of known sources and sinks could also bring top-down and bottom-up estimates closer together. For example, the general view of oxic soils as a sink for OCS has recently been challenged. Field (Maseyk et al., 2014; Billesbach et al., 2014) and incubation studies (Whelan et al., 2016) show that some oxic soils may shift from OCS uptake to emission depending on the temperature and water content. Furthermore, it has been speculated previously that vegetation uptake might not be solely responsible for the decrease in OCS mixing ratios in fall, because of the temporal lag between $CO_2$ and OCS minimum (Montzka et al., 2007). The observed seasonality in mixing ratios is a superposition of the seasonality of all individual sources and sinks. These seasonalities are currently neglected or associated with a considerable uncertainty. An improved understanding of the seasonality of the individual sources and sinks could help to better constrain the gap in the atmospheric budget. First steps to resolve OCS seasonality in sources and sinks are currently undertaken, e.g. in the case of anthropogenic emissions (Campbell et al., 2015).

All in all, better constraints on the seasonality and magnitude of the atmospheric OCS sources and sinks are critical for a better assessment of the role of this compound in climate and its application to quantify GPP on a global scale. This study confirms oceanic emission as the largest known single source of atmospheric OCS, but shows that its magnitude is unlikely to balance the gap in the atmospheric OCS budget.

## Appendix A: List of parameters

| Symbol | Meaning |
| --- | --- |
| $a_{350}$ | absorption coefficient of CDOM at 350nm |
| $a$ | fitted parameter in diurnal cycle of I |
| $b$ | fitted parameter in diurnal cycle of I |
| $c_{air}$ | concentration in air |
| $C_{OCS}$ | concentration of OCS in wtaer |
| $F$ | gas flux |
| $H$ | Henry constant |
| $I$ | downwelling solar radiation |
| $K$ | ion product of seawater |
| $k_w$ | water side transfer velocity in air-sea gas exchange |
| MLD | mixed layer depth |
| $p$ | photoproduction rate constant |
| SSS | sea surface salinity |
| SST | sea surface temperature |
| $Sc$ | Schmidt number |
| $t$ | time |
| $\theta$ | zenith angle |
| $u_{10}$ | wind speed at 10m height |
| UV | ultra violett radiation |
| $z$ | depth |

*Acknowledgements.* We thank the captain and crew of the research vessels SONNE I and II as well as HESPERIDES for assistance during the cruises SO235-OASIS (BMBF - FK03G0235A), SO243-ASTRA-OMZ (BMBF - FK03G0243A) and TransPEGASO. We thank H.W. Bange and A. Körtzinger for providing equipment for the continuous underway system and C. Schlundt for support during $CS_2$ measurements. This work was supported by the German Federal Ministry of Education and Research through the project ROMIC-THREAT (BMBF-FK01LG1217A and 01LG1217B) and ROMIC- SPITFIRE (BMBF- FKZ: 01LG1205C). Additional funding for C.A.M. and S.T.L. came from the Helmholtz Young Investigator Group of C.A. M., TRASE-EC (VH-NG-819), from the Helmholtz Association through the President's Initiative and Networking Fund and the GEOMAR Helmholtz-Zentrum für Ozeanforschung Kiel. K.K. acknowledges financial support from the EU FP7 StratoClim project (603557), and P.C. and R.S. acknowledge support from Spanish MINECO through PEGASO (CTM2012-37615). We gratefully acknowledge the data provided by ECMWF (ERAInterim) and NASA (MODIS Aqua). DKRZ and its scientific steering committee are gratefully acknowledged for providing the HPC and data archiving resources for this consortial project ESCiMo (Earth System Chemistry Integrated Modelling). E.A. acknowledges support from NASA Upper Atmosphere Research Program.

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

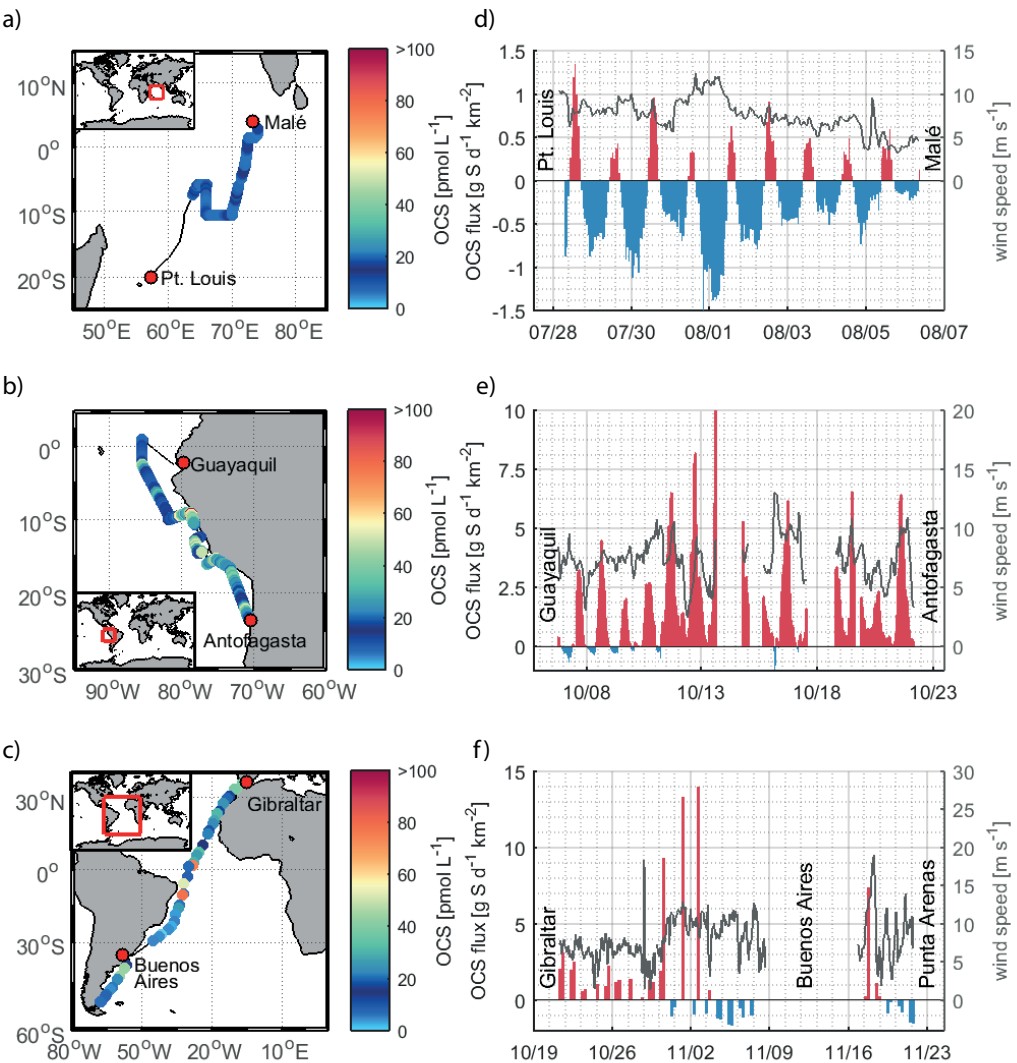

**Figure 1.** Observed OCS water concentrations and calculated emissions: Observations of OCS concentrations in the surface ocean during three cruises a) OASIS, b) ASTRA-OMZ, and c) TransPEGASO; and the corresponding emissions calculated based on the concentration gradient between water and marine boundary layer (d-f). Outgassing is indicated in red bars; oceanic uptake in blue bars. The grey line shows wind speed measured onboard the vessels. Flux data are shown with different scales on the y-axes. Data gaps occurred during stays in port and territorial waters or during instrument tests.

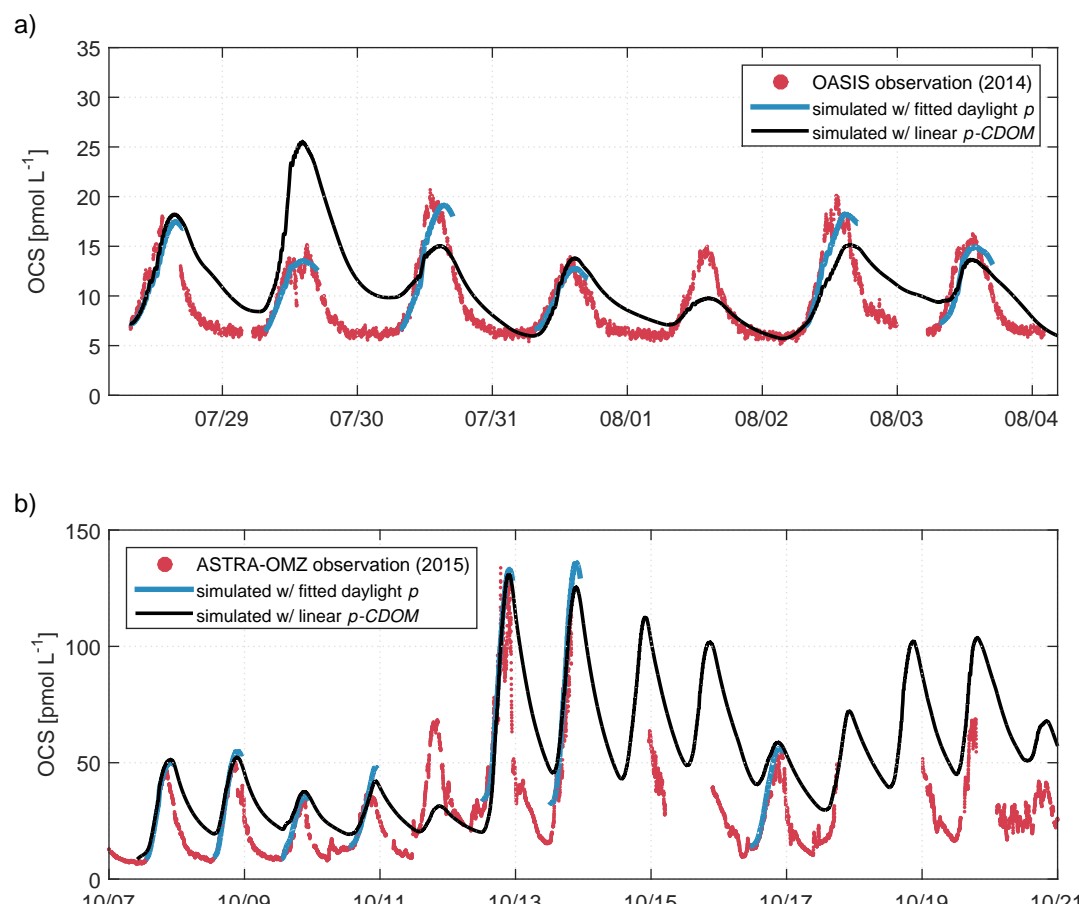

**Figure 2.** Box model simulations compared to observations: Comparison of simulated OCS water concentrations against measurements from the OASIS cruise to the Indian Ocean (a) and the eastern Pacific Ocean during the ASTRA-OMZ cruise (b). Blue indicates OCS concentrations with a least-square fit for the photoproduction rate constant $p$ during daylight, fitted individually for days with homogeneous water masses (SST, $a_{350}$). Black shows the simulation including the $p$ depending on $a_{350}$, obtained from linear regression of individually fitted $p$ with $a_{350}$ ($R^2$=0.71). The time on the x-axis is local time (GMT+5 during OASIS 2014, GMT-4 during ASTRA-OMZ 2015).

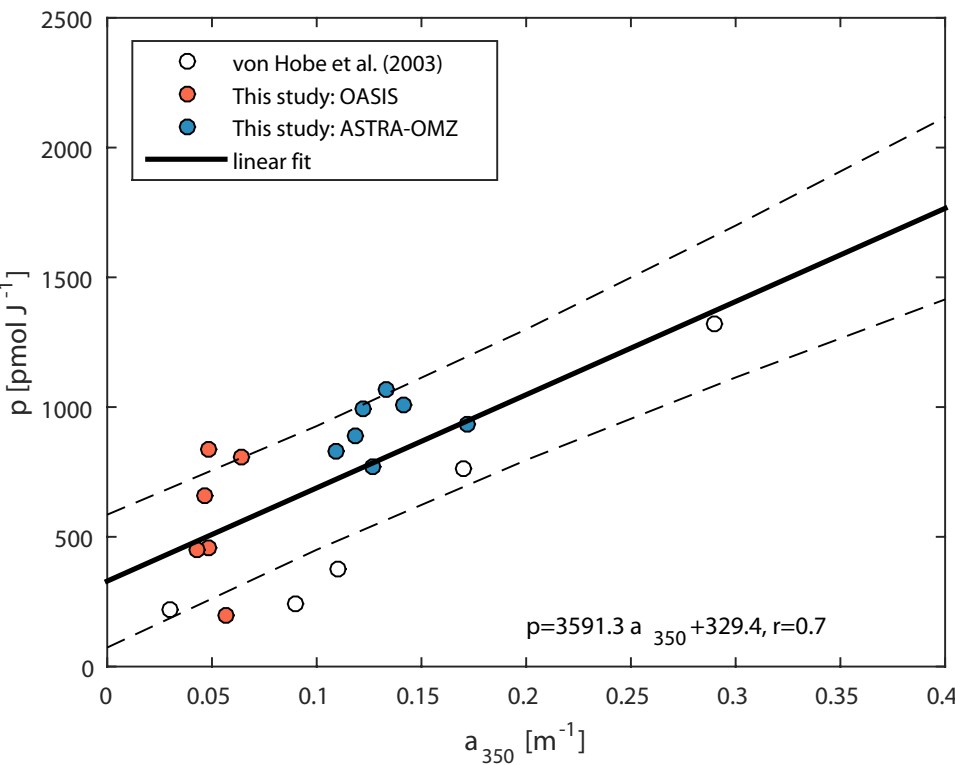

**Figure 3.** Dependence of photoproduction rate constant *p* on $a_{350}$ including own fits for *p* (resulting in blue lines in Fig. 2) and fits from a similar study (von Hobe et al., 2003). Dashed lines indicate the 95% confidence interval.

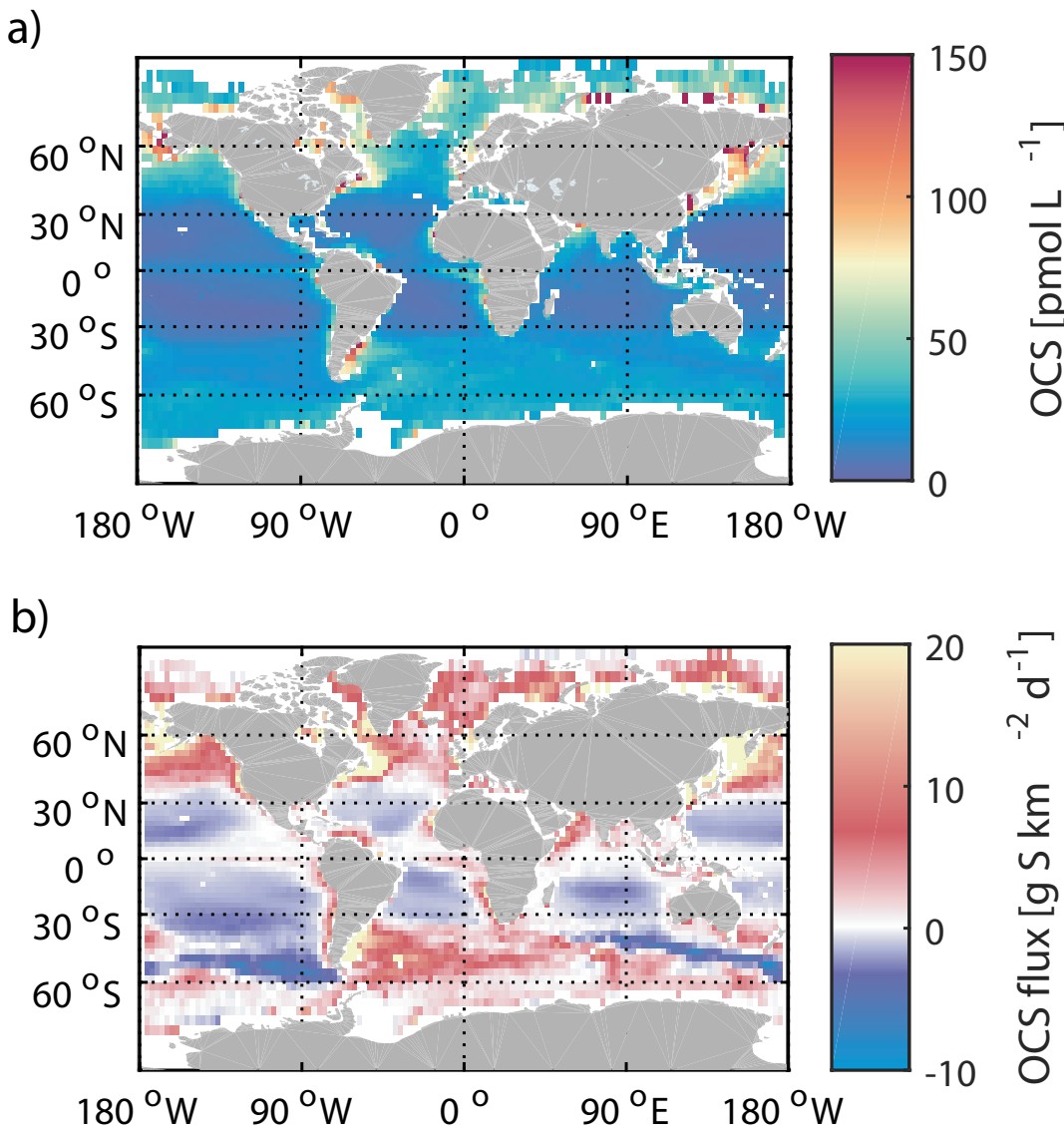

**Figure 4.** Annual mean of surface ocean concentrations of OCS simulated with the box model (a) and corresponding emissions (b).

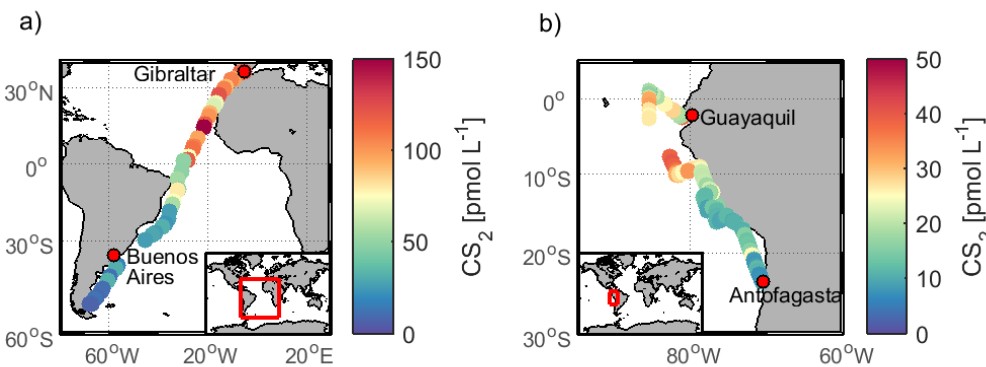

**Figure 5.** Measured concentration of $CS_2$ in surface waters during a) ASTRA-OMZ in the East Pacific Ocean and b) TransPEGASO in the Atlantic Ocean.

**Table 1.** Missing source estimates derived from top-down approaches: The listed studies used an increased vegetation sink and an *a priori* direct and indirect ocean flux to estimate the magnitude of the missing source. Assigning the missing source to oceanic emissions results in the total ocean flux listed here. Fluxes are given in Gg S per year.

| Reference | *a priori* ocean flux | missing source | total ocean flux |
|---|---|---|---|
| Suntharalingam et al. (2008) | 235 | 230 | 465 |
| Berry et al. (2013) | 276 | 600 | 876 |
| Kuai et al. (2015) | 289 | 800 | 1089 |
| Glatthor et al. (2015) | 276 | 714 | 992 |

**Table 2.** Global oceanic emission estimates of OCS: Direct ocean emission estimates of OCS from bottom-up approaches. Uncertainties are given in parenthesis as in the original paper either as range or ±standard deviation. ([a]units deviate from original paper, converted to Gg S for comparison, [b]upper limit)

| Reference | Emitted S as OCS (Gg S yr$^{-1}$) |
|---|---|
| **extrapolated from measurements** | |
| Rasmussen et al. (1982) | 320 ($\pm$160)[a] |
| Ferek and Andreae (1983) | 245[a] |
| Johnson and Harrison (1986) | 110-210[a] |
| Mihalopoulos et al. (1992) | 230 (110-210)[a] |
| Chin and Davis (1993) | 160 (85-340)[a] |
| Weiss et al. (1995b) | -16 (10-30)[a] |
| Ulshöfer and Andreae (1998) | 41-80[a] |
| Watts (2000) | 53 ($\pm$80)[a] |
| Xu et al. (2001) | 53[a] |
| **model simulations** | |
| Kettle (2002) | 41 ($\pm$154) |
| Launois et al. (2015a) | 813 (573-3997) |
| **This study** | **130**[b] |

**Table 3.** Average, standard deviation and range of parameters observed during the cruises OASIS (Indian Ocean, 2014), ASTRA-OMZ (Pacific Ocean, 2015) and TransPEGASO (Atlantic Ocean, 2014).

| | | average (± std. dev.) | minimum | maximum |
|---|---|---|---|---|
| **OASIS (Indian Ocean)** | | | | |
| OCS sea surface concentration | [pmol L$^{-1}$] | 9.1 (±3.5) | 5.1 | 20.7 |
| OCS flux | [g S d$^{-1}$ km$^{-2}$] | -0.25 (±0.5) | -1.6 | 1.5 |
| SST | [°C] | 27.0 (±1.4) | 22.2 | 32.0 |
| salinity | [-] | 34.9 (±0.3) | 34.3 | 35.4 |
| wind speed | [m s$^{-1}$] | 7.6 (±2.1) | 0.2 | 14.5 |
| a$_{CDOM}$(350) | [m$^{-1}$] | 0.03(±0.02) | n.d. | 0.12 |
| **ASTRA-OMZ (Pacific Ocean)** | | | | |
| OCS sea surface concentration | [pmol L$^{-1}$] | 28.3 (±19.7) | 6.5 | 133.8 |
| OCS flux | [g S d$^{-1}$ km$^{-2}$ | 1.5 (±2.1) | -1.5 | 19.9 |
| CS$_2$ sea surface concentration | [pmol L$^{-1}$] | 17.8 (±8.9) | 6.7 | 40.1 |
| CS$_2$ flux | [g S d$^{-1}$ km$^{-2}$] | 4.1 (±3.2) | 0.2 | 14.4 |
| SST | [°C] | 20.1 (±2.9) | 15.6 | 26.9 |
| salinity | [-] | 35.0 (±0.43) | 33.4 | 35.5 |
| wind speed | [m s$^{-1}$] | 7.4 (± 2.0) | 0.3 | 15.5 |
| a$_{CDOM}$(350) | [m$^{-1}$] | 0.15 (±0.03) | 0.1 | 0.24 |
| **TransPEGASO (Atlantic Ocean)** | | | | |
| OCS sea surface concentration | [pmol L$^{-1}$] | 23.6 (±19.3) | 2.6 | 78.3 |
| OCS flux | [g S d$^{-1}$] | 1.3 (±3.5) | -1.7 | 14.0 |
| CS$_2$ sea surface concentration | [pmol L$^{-1}$] | 62.5 (±42.1) | 23.2 | 154.8 |
| CS$_2$ flux | [g S d$^{-1}$ km$^{-2}$] | 13.7 (±9.8) | 0.3 | 33.9 |
| SST | [°C] | 22.6 (±6.3) | 7.1 | 29.6 |
| salinity | [-] | 34.9 (±2.6) | 28.4 | 38.1 |
| wind speed | [m s$^{-1}$] | 7.4 (±3.1) | 0.4 | 19.0 |
| a$_{CDOM}$(350) | [m$^{-1}$] | 0.13 (±0.11) | 0.0023 | 0.45 |

**Table 4.** Comparison of water properties relevant for OCS production and consumption for the cruises OASIS (Indian Ocean, Jul-Aug 2014) and ASTRA-OMZ (Östlicher Pazifik, Oct-Nov 2015) with the assumed source region in the Pacific warm pool (15°N-15°S, 120°-180°E). Data from cruises are in situ-measurements; the data for the Pacific warm pool was extracted from climatological monthly means from sources for the global model run as specified in the supplementary material.*calculated from an annual mean diurnal cycle based on ERA-Interim sunshine duration and flux. SR=surface radiation, daily integral **assumed pH=8.15 for box model simulation.

| Parameter | OASIS | ASTRA-OMZ | Pacific Warm Pool |
|---|---|---|---|
| SST [°C] | $27.0\pm1.0$ | $19.6\pm2.6$ | $28.9\pm0.9$ |
| SSS [g kg$^{-1}$] | $35.0\pm0.3$ | $35.1\pm0.3$ | $34.5\pm0.42$ |
| wind speed [m s$^{-1}$] | $8.2\pm1.7$ | $7.5\pm1.8$ | $5.3\pm0.4$ |
| a$_{350}$ [m$^{-1}$] | $0.039\pm0.02$ | $0.146\pm0.02$ | $0.050\pm0.08$ |
| I [W m$^{-2}$] | $226.5\pm303.0$ | $196.4\pm283.1$ | $206.4\pm286.6$* |
| SR [J m$^{-2}$] | $1.9\ 10^7\pm1.7\ 10^6$ | $1.6\ 10^7\pm4.5\ 10^6$ | $8.9\ 10^6\pm1.3\ 10^6$ |
| pH [-] | $8.03\pm0.01$ | -** | $8.07\pm0.01$ |
| MLD [m] | $43.3\pm15.8$ | $18.9\pm7.5$ | $35.9\pm14.1$ |

**Table 5.** Comparison of previous shipcampaign measurements with corresponding month and approx. geolocation from the global box model in this study (L2016), taken either from figures or tables as provided in the original references. Note that the box model is based on input data from climatological means that do not fully represent the conditions encountered during the respective cruises. Only observational data with measurements of the full diel cycle were included for comparison. n=number of measurements. *converted from ng $L^{-1}$ with a molar mass of OCS of 60.07g. **converted from ng S $L^{-1}$ with a molar mass of S of 32.1g.

| References | Season | Region | Mean OCS±std. [pmol $L^{-1}$] | n | L2016 mean [pmol $L^{-1}$] |
|---|---|---|---|---|---|
| Mihalopoulos et al. (1992) | | open Indian Ocean 20°N-37°S | | | |
| | Mar/May 1986 | OCEAT II | 19.9±0.5* | 20 | 11.2±6.3 |
| | Jul 1987 | OCEAT III | 19.9±1.0* | 14 | 17.7±13.1 |
| Staubes and Georgii (1993) | Nov-Dec 1990 | Wedell Sea 40°-72°S,72°W-24°E | 109** | 126 | 66.6±49.8 |
| Ulshöfer et al. (1995) | | North Atlantic Ocean | | | |
| | Apr/May 1992 | 47°N 20°W | 14.9±6.9 | 118 | 42.8±11.3 |
| | Jan 1994 | 48-50°N, 10-17°W | 5.3±1.6 | 120 | 8.9±3.2 |
| | Sep 1994 | 48-50°N, 10-17°W | 19.0±8.3 | 23 5 | 33.4±3.5 |
| Flöck and Andreae (1996) | Jan 1994 | Northeast Atlantic Ocean 49°N, 12°W | 6.7 (4-11) | 120 | 9.6±3.7 |
| Ulshöfer and Andreae (1998) | Mar 1995 | West Atlantic 32°N, 64°W | 8.1±7.0 | 323 | 15.8 |
| von Hobe et al. (1999) | Jun/Jul 1997 | Northeast Atlantic Ocean 30-40°N, 8-15°W | 23.6±16.0 | 940 | 30.5±12.6 |
| Kettle et al. (2001) | Sep/Oct 1998 | Atlantic transect 50°N-60°S, 1°-64°W | 21.7±19.1 | 783 | 22.9±3.2 |
| von Hobe et al. (2001) | Aug 1999 | Sargasso Sea/BATS 32°N, 64°W | 8.6±2.8 | 518 | 8.1 |
| Xu et al. (2001) | Oct/Nov 1997 | Atlantic meridional transect 53°N-34°S,25°W-20°E | 14.8±11.4 | 306 | 11.8±12.7 |
| | May/Jun 1998 | Atlantic meridional transect 53°N-34°S,25°W-20°E | 18.1±16.1 | 440 | 27.8±47.9 |