# Peer review of "Direct oceanic emissions unlikely to account for the missing source of atmospheric carbonyl sulfide"

_Atmospheric Chemistry and Physics, 2016_

## Referee Comment (RC1) · Anonymous Referee #1 · 8 Oct 2016

Overview:

The authors present new bottom-up measurements and analysis of COS and CS2 from 3 ocean cruises. The ocean source is a dominant source of uncertainty in global COS budgets so the authors should be commended for presenting new, high quality data. However, the central conclusion in the manuscript title and text is not supported by the measurements. Nevertheless, the measurements and analysis provide a very important contribution to understanding COS budgets and I suggest only simple, but critical, revisions to the title and text.

Specific Comments:

[Figure]

The title and several statements in the text should be changed so that the conclusions become consistent with the data. In particular, the measurements are not a representative sample for extrapolating to the global source and thus conclusions on the global source should not be made. There are of course many other exciting conclusions that are possible. The measurements are not representative of the global source for following reason. Global satellite observations show global hot spot for the source in the Pacific Warm Pool for most of the year and in June/July/Aug more broadly across the tropical and mid-latitude Pacific. The cruise measurements from this study that are used in the global extrapolation do not cover this critical region. If the author's were looking to uncover information on the missing source they should target locations/times that top-down data points to for the missing source. However, the cruise data presented here are in times and locations were the top-down data suggest that the ocean source should be small or even a sink. I still think the global analysis is useful to include because it is already done and likely points to the problem with scaling up from non-representative data.

Robust conclusions for this study could instead focus on describing the ocean source for the times/locations of the three cruises shown in Figure 1. A qualitative comparison could also be made with previous top-down analysis. This seems to be good ground for an exciting conclusion of consistency between top-down and bottom-up estimates. In this case there appear to be some strong similarities between the bottom -up and top-down estimates. The TranPEGASO cruise covers a section of the Atlantic in Oct/Nov, showing a small source. This is qualitatively consistent with the MIPAS data along the same path in Sept/Oct/Nov. The Oasis cruise covers a small area in the central Indian Ocean in July/Aug showing a sink. This is roughly consistent with a MIPAS Jun/Jul/Aug map and a TES June map that show this region to be on the border between a source and sink. ASTRA-OMZ show a strong source in October for the Peru-Chile upwelling region. MIPAS Sept/Oct/Nov doe not show this. However, MIPAS is an upper troposphere estimate so it is not expected to provide a close relationship to surface fluxes in regions without strong atmospheric convection such as the Peru-Chile upwelling region. TES provides a lower altitude sensitivity and could provide a better top-down on small regions of sources such as the Peru-Chile upwelling regions. While TES data have only been published for June, TES retrievals for other months are in preparation.

Several revisions are needed in the introduction. Page 2 indicates that top-down studies were not consistent with the Kettle bottom-up estimates for sources and sinks. This should be corrected to say that the bottom-up and top-down info does not agree with Kettle. Kettle was a misinterpretation of the bottomup information from plant studies which was first pointed out in the bottom-up study of Sandoval-Soto et al. and then later confrined by multiple topdown studies (Campbell et al, Sunthralingam et al, Berry et al, etc.) and other bottomup studies using chamber (Stimler et al) and canopy (Asaf et al, Maseyk et al) approaches.

The top down evidence from the global scale should be better specified. First it should be clear that there are four independent lines of five independent lines of evidence that point to a tropical source: MIPAS satellite (Glatthor et al), TES satellite (Kuai et al), FTIR (Wang et al, ACP, 2016... this ref isn't in the manuscript but might be added), NOAA and HIPPO observations (Berry et al, Kuai et al, Suntharlingam et al).

A critical point should be raised to alert the reader to alternative explanations for the top-down trends. In particular, the MIPAS remote sensing data is the upper troposphere (∼10km) and transport from Asia to the upper troposphere in the deep tropics (e.g. Ashfold et al ACP 2015). Recent anthropogenic emission estimates for Asia are not yet sufficient to explain the missing source but they are based on very little bottom-up data from Asia (Campbell et al 2015). Other hypotheses could be mentioned as well such as a soil source which has been shown in a recent survey of global soils but not particularly large in the tropics (Whelan et al ACP 2016). Biomass burning is another but the most recent review of emission factors shows a relatively small source (Campbell et al 2015).

The Van Hobe study was included but more could be done to explain what other cruise

data is available. The introduction needs to explain how the cruise measurements and ocean box modeling fit within the context of previous cruise measurement and ocean box modeling. Were these cruises in seasons or locations that have others have not gone?

The introduction or discussion could also compare the modeling approach here to what has been done previously. In particular the recent paper by Launois et al.

Some comments may be helpful on alternative approaches for validating these flux estimates. Spatial gradients in atmospheric mixing ratios have been used recently (Berkelhammer et al below). Are other approaches also possible? M. Berkelhammer, H.C. Steen-Larsen, A. Cosgrove, A. Peters, R. Johnson, M. Hayden and S.A. Montzka (Minor Revisions, July 2016) Radiation and atmospheric circulation controls on carbonyl sulfide concentrations in the marine boundary layer. Journal of Geophysical Research (available upon request).

Section 2.3 should describe how the box model relates to the measurements. This is done in the results section "Following an earlier study (von Hobe et al., 2003), we use our observations ..." but belongs in the methods. A few additional sentences of explanation may be helpful. Why was the parameter p chosen for fitting the model as opposed to the numerous other parameters. Were other parameters also examined? If not then perhaps this should be stated as an important next step for future work. Why was the von Hobe et al., 2003 study used but not other studies? What is the spatial and temporal extent of the Von Hobe data?

"global radiation I was" not sure what "I" is

page 6, explain what you mean by "case study simulations"

define "CTD profiles"

The methods section should also include a summary of the time and location of the 3 cruises.

The section "2.1 Measurement set-up for trace gases" present a different method for each cruise. It would be helpful if this section also summarized the impact of having different methods on the different cruises in terms of different precision and other factors that may or may not influence the quality of these measurements.

Table 3 missing km^-2 in TransPEGASO flux

"an non-negligible" to "a"

Some description is needed of the error associated with assuming a constant atmospheric mixing ratio on TransPEGASO. Seasonal and spatial variation in atmospheric mixing ratios can be on the order of 100 ppt.

"leaving the missing source still explained" should be "unexplained"?

Again this is an important contribution of new, high quality data and a well written manuscript. The authors present a compelling approach and with further data could provide a key to closing the global COS budget.

---

## Referee Comment (RC2) · Anonymous Referee #2 · 19 Oct 2016

I find it striking that such broad generalizations are made from data that cover a fairly small portion of the global ocean. The authors have a great deal of new observations to address the important issue of quantifying OCS fluxes from the ocean. But to draw broad conclusions without considering the previous data more carefully is inappropriate. Their data add to the picture in useful and interesting ways. To "conclusively answer the question of whether the missing OCS source... can really be ascribed to the direct OCS emissions from tropical oceans" would seem to require another level of effort that isn't yet part of this manuscript.

I realize that the model helps the authors extrapolate their results to broader scales, but the results derived are only as good as the data considered by the model. If the

authors really hope to be able to "constrain the variability of OCS emissions in the tropic[al ocean]" then I would think they would have to consider the details of previous ocean-going measurements (ocean basin, ocean regime, season, etc.) together with their new data to determine if, in fact, that most of the major global ocean regimes have been adequately sampled to allow such a conclusion. For example, it is informative and important that their results are consistent with an upper flux limit from Kettle et al., 2002, but no mention is made of how that consistency actually increases our understanding of total OCS flux from the ocean. If the sampled regions were similar, then this is a confirmation of that original estimate, but potentially not much progress in understanding the broader-scale contribution of the ocean to atmospheric OCS abundances. Suggestion: scale back the broad conclusions and focus on your results and how they compare with others, or consider the broader literature on OCS flux measurements (observation-based and model-derived) in a more detailed fashion to determine if the available data allow an accurate quantification of the global and, most importantly, tropical OCS flux from the ocean.

On uncertainties. Any comparison of derived oceanic fluxes with a shortfall needs to fully consider uncertainties. Many uncertainties are discussed (air-sea exchange, mixed layer depth, parameterization of production, etc.,) but aren't explicitly included in the derivation of the direct flux of 130 GgS/yr and in the discussion of the global budget discrepancy (no uncertainty is provided on this number). Similarly, the origin of stated uncertainties in the derivation of indirect fluxes from DMS are not discussed, but I would imagine are substantially larger then estimated. Without a fair treatment of these uncertainties, it isn't possible to gauge the true magnitude of the budget shortfall, which is a main point of the manuscript.

Details: P1, line 7-8. It only has implications for GPP derived from OCS observations on certain scales, not all.

p.2, line 25-26. It also makes much more chemical sense given our understanding of how COS interacts with carbonic anhydrase in leaf waters.

Table 1 and p. 9 line 27-30. The paper from Suntharalingam does not state or suggest that the missing source is oceanic.

p.3 line 13. Underrepresentation of global flux is also true for the measurements made here, despite the use of a box model for extrapolation.

Figure 3, and line 10, p. 9. Is R = 0.7 or R^2 = 0.7 in this relationship? These different values seem given for one relationship. Also, given the non-normal distribution of these results this value is not as significant as one might presume.

The conclusion section is a bit unusual in that it includes speculations not supported by the work. It also reads like a research planning document. On what basis do the authors conclude (p. 11, line 22) that observations could be reproduced "without increasing the vegetation sink"? An extensive body of recent literature has suggested that the interaction between OCS and vegetation is best explained by a substantially larger sink than discussed in earlier papers; to discount those studies without substantial evidence is inappropriate, making this conclusion one that does not follow from the evidence presented in the paper.

---

## Short Comment (SC1) · 28 Oct 2016

The bottom-up approach of COS concentrations in the oceanic mixed layer used in this study relies on the relationship shown in Fig. 3 where the photoproduction rate constant (or quantum yield) at 350 nm (p350) is linearly correlated to the CDOM absorption coefficient at 350 nm. A similar study was carried out by von Hobe et al. (2003) but, in that study, the authors investigated p313 and a313. A wavelength of 313 nm was used because "it falls into the center of the wavelength band for UV and coincides well with the wavelength where COS surface production rate spectra show a maximum in tropical waters (Weiss et al., 1995)." I wonder why the authors calculated a new relationship at 350 nm which is the wavelength where COS surface production rate

spectra show values about twice lower than at 313 nm (see Fig. 2C of Weiss et al. (1995)). Can you clarify this point?

The average action spectrum of COS used by von Hobe et al. (2003) is the one established by Weiss et al. (1995) from incubation experiments carried out in tropical and Antarctic regions. Weiss et al. compared the average action spectrum from their work to that of Zepp and Andreae (1990) who investigated coastal North Sea water samples. Weiss et al. noted that differences in DOM origin between the two works may explain the pronounced discrepancy in relative action spectra. My interpretation of data shown in Fig. 3 is that the quantum yield of COS is about twice higher in the surface waters of the Indian and Pacific oceans than in the Atlantic. Because the COS quantum yield exhibits quite large variations even far from coastal areas, I wonder how the Lennartz et al.'s relationship can be representative for the global ocean.

The latitude-time plots in Fig. S5 (this work) and in Fig. 8 of Launois et al. (2015) look pretty much the same in terms of hydrolysis rates in tropical regions (1 to 2 pmol L-1 h-1). How is this possible knowing that your model predicts considerably lower COS levels in tropical waters?

My last concern, to conclude, relates to the lack of validation of your extrapolations from published inventories. I refer to that of Mihalopoulos et al. (1992) who gathered cruise observations of the supersaturation ratio (SR) of COS in coastal areas and in the open ocean (their Tables 1 & 2) and of the latitudinal variations of seawater COS concentrations (their Fig. 2). The calculation of oceanic fluxes requires 6 variables (kw, T, wind speed, Cw, Ca and H). That of SR only requires 4 variables (Cw, Ca, T and H). The direction of the flux can be assessed from SR values (SR<1 = sink; SR>1 = source). I think you should try to evaluate simulated monthly maps of SR against the inventories of Mihalopoulos et al. (1992). Scanned data plots can be digitized using on-line facilities.

Launois, T., S. Belviso, L. Bopp, C.G. Fichot, and P. Peylin. A new model for the global

biogeochemical cycle of carbonyl sulfide – Part 1: Assessment of direct marine emissions with an oceanic general circulation and biogeochemical model. Atmos. Chem. Phys., 15, 2295-2312, 2015.

Mihalopoulos, N., B.C. Nguyen, J.P. Putand, and S. Belviso. The oceanic source of carbonyl sulfide (COS). Atmos. Environ. 26A, 8, 1383-1394, 1992.

Von Hobe M., R.G. Najjar, A.J. Kettle, and M.O. Andreae. Photochemical and physical modeling of carbonyl sulfide in the ocean. J. Geophys. Res. 108, C7, 3229, doi:10.1029/2000JC000712, 2003.

Weiss, P. S., S.S. Andrews, J.E. Johnson, and O.C. Zafiriou. Photoproduction of carbonyl sulfide in south Pacific Ocean waters as a function of irradiation wavelength. Geophys. Res. Lett. 22, 3, 215-218, 1995.

Zepp, P.S., and M.O. Andreae. Photosensitized formation of carbonyl sulfide in sea water. In Effects of solar ultraviolet radiation on biogeochemical dynamics in aquatic environments. Blough and Zepp Eds., WHOI technical report 90-09, 1990.

---

## Short Comment (SC2) · 9 Nov 2016

Dear Sinnika,

I read your paper with great interest and I have a few comments that relate primarily to the photoproduction part of your work. Note I generally support the reviewers' comments on better acknowledging and including uncertainties associated with your estimates. This is especially important because the paper's conclusions contradict the main findings of other published work

In particular:

1. The photoproduction rate calculations deserve to be explained in more details, especially when it comes to the global calculations. In particular, are the rates reported integrated over the mixed layer (as is suggested by equation 2) or are they an average for the mixed layer, or at a discrete depth (as would be suggested by the pmol L-1 h-1 units reported on figure S5)? The caption mentions production rates but that's it. f the rates are integrated over the mixed layer then the units of photoproduction rates should be in units of pmol per unit time and unit AREA (not per unit VOLUME). It would be good to clarify this because it would help facilitate comparisons with other approaches. Note I attached (see supplement) a number of figures of calculated rates using the Fichot and Miller (2010) model implemented with the single wavelength-resolved apparent quantum yield of Weiss et al. (1995) for comparison purposes and to illustrate the differences in photochemical rates calculated at different depths or for different depth ranges (mixed layer, and sunlit layer).

2. The attempt to constrain the variability in the photochemical rate constant p is welcomed, but it should also be recognized in the paper that using wavelength-independent p for the global calculations can also lead to significant and potentially large uncertainties in the calculated rate. In the van Hobe (2013) paper, it was suggested this had a little impact, but I believe the comparison using wavelength-independent and wavelength-resolved constants was made at the same location where there should be minor differences in the spectral characteristics of the UV downwelling irradiance spectrum. For global calculations, this simplification can be much more problematic because the spectral characteristics of the downwelling irradiance can vary quite dramatically with latitude or with atmospheric conditions. This is potentially a large source of uncertainty that would be good to discuss in the paper.

3. It should also be clearly acknowledged that the modeled downwelling (or downward) irradiance (as described in the paper) are crude approximations and do not even include the effects of clouds for example. This is another important source of uncertainty.

4. It should also be better mentioned that there are uncertainties in the a350 retrieved from satellite data and that uncertainty will be compounded when p is derived from

a350. There is also uncertainty due to the fact that the GSM model used to retrieved the absorption by detritus+CDOM at 443 nm and not just CDOM. Finally, another uncertainty is associated with the use of a single spectral slope for CDOM to derive a350 from a443.

All these combined uncertainties can amount to a large error in the estimates of OCS photoproduction alone, and I think there is a need to better acknowledge all sources of uncertainty and their implications on the conclusions, and there is a need for a more cautious language with regards to the conclusions of this work.

I also would like to take the opportunity to use the attached figures of COS photoproduction rates integrated over the mixed layer to highlight that the regions studied in your paper are generally not as photochemically active (generally lower photoproduction rates) as the region studied in Mihalopoulos et al. (1992). This is in relation to the comment posted by Dr. Belviso.

I hope this information is helpful.

Best,

Cedric

Please also note the supplement to this comment:
http://www.atmos-chem-phys-discuss.net/acp-2016-778/acp-2016-778-SC2-supplement.pdf

**Supplement:**

JULY

Integrated from 0- to depth of MLD

COS photoproduction rate  (mol COS m$^{-2}$ day$^{-1}$)

OCTOBER

COS photoproduction rate  (mol COS m$^{-2}$ day$^{-1}$)

——————— Cruise tracks

**JULY**

COS photoproduction rate (mol COS m$^{-2}$ day$^{-1}$)

Integrated from 0- to depth of max UV penetration

**OCTOBER**

COS photoproduction rate (mol COS m$^{-2}$ day$^{-1}$)

[Figure]

COS photoproduction rate at Xm depth (pmol COS L$^{-1}$ h$^{-1}$)

**JULY**

[Figure]

COS photoproduction rate MLD average (pmol COS L$^{-1}$ h$^{-1}$)

**OCTOBER**

[Figure]

[Figure]

COS photoproduction rate MLD average (pmol COS L$^{-1}$ h$^{-1}$)

[Figure]

COS photoproduction at 0m depth (pmol L$^{-1}$ h$^{-1}$)

COS photoproduction at 1m depth (pmol L$^{-1}$ h$^{-1}$)

COS photoproduction at 2m depth (pmol L$^{-1}$ h$^{-1}$)

COS photoproduction at 5m depth (pmol L$^{-1}$ h$^{-1}$)

COS photoproduction average MLD (pmol L$^{-1}$ h$^{-1}$)

---

## Author Comment (AC1) · 22 Nov 2016

We thank the reviewer#1 for the very constructive and detailed review that helps to clarify and strengthen our argumentation. In the following, we address the raised points directly, with the *review in italics* and **our reply in bold** font.

*Overview:*
*The authors present new bottom-up measurements and analysis of COS and CS2 from 3 ocean cruises. The ocean source is a dominant source of uncertainty in global COS budgets so the authors should be commended for presenting new, high quality data. However, the central conclusion in the manuscript title and text is not supported by the measurements. Nevertheless, the measurements and analysis provide a very important contribution to understanding COS budgets and I suggest only simple, but critical, revisions to the title and text.*

**We will address the conclusions drawn in the specific comment below, but changed the title to "Direct oceanic emissions are unlikely to account for the missing source", because our observations, previous observations and the box model reproducing both reasonably well reveal a direct emission estimate that is a factor of 3-8 below the missing source estimate and thus very unlikely. We still deem indirect emissions as unlikely to account for the whole missing source, but acknowledge the uncertainty related to these emission estimates by changing the title.**

*Specific Comments:*

*The title and several statements in the text should be changed so that the conclusions become consistent with the data. In particular, the measurements are not a representative sample for extrapolating to the global source and thus conclusions on the global source should not be made. There are of course many other exciting conclusions that are possible. The measurements are not representative of the global source for following reason. Global satellite observations show global hot spot for the source in the Pacific Warm Pool for most of the year and in June/July/Aug more broadly across the tropical and mid-latitude Pacific. The cruise measurements from this study that are used in the global extrapolation do not cover this critical region. If the author's were looking to uncover information on the missing source they should target locations/times that top-down data points to for the missing source. However, the cruise data presented here are in times and locations were the top-down data suggest that the ocean source should be small or even a sink. I still think the global analysis is useful to include because it is already done and likely points to the problem with scaling up from non-representative data.*

**Obviously, the use of the word "extrapolation" in the context of relating our new observations to the global source estimate has been misleading. To estimate tropical and global oceanic OCS emissions, we use a model that is much simplified with respect to mixed layer dynamics and vertical mixing, but contains state-of-the-art parameterizations of the known processes governing OCS concentrations in seawater. To our knowledge, these process parameterizations have not been seriously challenged and have always yielded good results in studies comparing observed and simulated OCS for individual cruises from different parts of the oceans. In our paper, we present the first extensive study of this type focused on the tropical ocean, and find that here, too, the established model reproduces observed OCS quite well. The tropical observations are used to fine-tune the model, in particular corroborating the relationship of photochemical production on $a_{CDOM}$. The model is then used in a similar way as has been done by Kettle (2002) to estimate fluxes from all regions of the global ocean and integrate the results to yield a global flux estimate.**

**To clarify this approach and its benefits, we apply the following changes in a revised manuscript:**
1. **We add a more thorough description of the development and history of the box model and then point out how the new observations are used to fine-tune the model and to enlarge the range of biogeochemical regimes, for which the model is tested.**

**p. 6, l. 23ff: "A box model to simulate surface concentration of OCS is further developed from the latest version from von Hobe et al. (2003, termed vH2003), where concentrations along the cruistracks of 5 Atlantic cruises have been simulated and compared. The vH2003 model results from successful tests and validation to observations on several cruises to the Altantic Ocean covering all seasons (i.e. Flöck and Andreae (1996) in January 1994, Uher and Andreae (1997) in April/May 1992, Von Hobe et al. (1999) in June/July 1997, Kettle et al. (2001) in September/October 1998). By comparing photoproduction rate constants of the 5 cruises to CDOM absorption, von Hobe (2003) suggests a second order process for photoproduction with the photoproduction rate constant being dependent on the absorption of CDOM in seawater.**

**In our approach, we test vH2003 along the cruise track of two cruises, include a new way of determining the photoproduction rate constant (see below) and apply it with global climatological input (termed L2016). (Kettle, 2000; Kettle, 2002) applied a similar version of vH2003 globally, which included an optimized photoproduction constant from Atlantic transect cruise data, an optimized constant light-independant production and a linear regression to obtain CDOM from chlorophyll a. In comparison to K2000, we use (i) a new way of determining the photoproduction rate constant incorporating information from three ocean basins, (ii) the most recent parameterization of light-independent production available, and (iii) satellite observations for sea surface CDOM instead of an empiric relationship based on chlorophyll a.**

**Launois et al. (2015) implemented parameterizations for light-independant production, hydrolysis and air-sea exchange similar to vH2003 in the 3D global ocean model NEMO-PISCES. The main differences to the approach used here is the lack of accounting for mixing in L2016 (discussed in section 3.2.2., which will theoretically lead to higher simulated concentrations in our case) and the application of a photoproduction rate constant in our model that incorporates information from three open ocean basins in contrast to one from a study in the North Sea (Launois et al., 2015)."**

2. In a new table (Tab. 4), we compare the physico-chemical conditions encountered during our OASIS and ASTRA-OMZ to those of the Pacific Warm Pool where the atmospheric inversion studies suggest the hotspot of oceanic OCS emissions, demonstrating that the conditions in the Indian Ocean and the Pacific Warm Pool are similar and the known processes should thus yield similar OCS concentrations and direct fluxes.

| Parameter | OASIS | ASTRA-OMZ | Pacific Warm Pool |
|---|---|---|---|
| SST [°C] | $27.0\pm1.0$ | $19.6\pm2.6$ | $28.9\pm0.9$ |
| SSS [g kg$^{-1}$] | $35.0\pm0.3$ | $35.1\pm0.3$ | $34.5\pm0.42$ |
| wind speed [m s$^{-1}$] | $8.2\pm1.7$ | $7.5\pm1.8$ | $5.3\pm0.4$ |
| a$_{350}$ [m$^{-1}$] | $0.039\pm0.02$ | $0.146\pm0.02$ | $0.050\pm0.08$ |
| I [W m$^{-2}$] | $226.5\pm303.0$ | $196.4\pm283.1$ | $206.4\pm286.6$* |
| SR [J m$^{-2}$] | $1.9\ 10^7\pm1.7\ 10^6$ | $1.6\ 10^7\pm4.5\ 10^6$ | $8.9\ 10^6\pm1.3\ 10^6$ |
| pH [-] | $8.03\pm0.01$ | -** | $8.07\pm0.01$ |
| MLD [m] | $43.3\pm15.8$ | $18.9\pm7.5$ | $35.9\pm14.1$ |

3. In another table (Tab. 5), we compare the results of our global box model simulations to previously observed concentrations and fluxes.

| References | Season | Region | Mean OCS±std. [pmol L$^{-1}$] | n | L2016 mean [pmol L$^{-1}$] |
|---|---|---|---|---|---|
| Mihalopoulos et al. (1992) | | open Indian Ocean 20°N-37°S | | | |
| | Mar/May 1986 | OCEAT II | 19.9±0.5* | 20 | 11.2±6.3 |
| | Jul 1987 | OCEAT III | 19.9±1.0* | 14 | 17.7±13.1 |
| Staubes and Georgii (1993) | Nov-Dec 1990 | Wedell Sea 40°-72°S,72°W-24°E | 109** | 126 | 66.6±49.8 |
| Ulshöfer et al. (1995) | | North Atlantic Ocean | | | |
| | Apr/May 1992 | 47°N 20°W | 14.9±6.9 | 118 | 42.8±11.3 |
| | Jan 1994 | 48-50°N, 10-17°W | 5.3±1.6 | 120 | 8.9±3.2 |
| | Sep 1994 | 48-50°N, 10-17°W | 19.0±8.3 | 23 5 | 33.4±3.5 |
| Flöck and Andreae (1996) | Jan 1994 | Northeast Atlantic Ocean 49°N, 12°W | 6.7 (4-11) | 120 | 9.6±3.7 |
| Ulshöfer and Andreae (1998) | Mar 1995 | West Atlantic 32°N, 64°W | 8.1±7.0 | 323 | 15.8 |
| von Hobe et al. (1999) | Jun/Jul 1997 | Northeast Atlantic Ocean 30-40°N, 8-15°W | 23.6±16.0 | 940 | 30.5±12.6 |
| Kettle et al. (2001) | Sep/Oct 1998 | Atlantic transect 50°N-60°S, 1°-64°W | 21.7±19.1 | 783 | 22.9±3.2 |
| von Hobe et al. (2001) | Aug 1999 | Sargasso Sea/BATS 32°N, 64°W | 8.6±2.8 | 518 | 8.1 |
| Xu et al. (2001) | Oct/Nov 1997 | Atlantic meridional transect 53°N-34°S,25°W-20°E | 14.8±11.4 | 306 | 11.8±12.7 |
| | May/Jun 1998 | Atlantic meridional transect 53°N-34°S,25°W-20°E | 18.1±16.1 | 440 | 27.8±47.9 |

To discuss this table and compare to previous shipboard measurements, we added a new section "3.2.1. Comparison to previous shipbased measurements" p. 11, l. 18:

"The global simulation of OCS surface water concentrations generally reproduced the lower picomolar range of concentrations (Tab. 5), the seasonal pattern of higher concentrations during summer compared to winter (as e.g. in (Ulshöfer et al., 1995)) and the spatial pattern of higher concentrations in higher latitudes (e.g. Southern Ocean, (Staubes and Geogrii, 1993)). Given that monthly means of a model simulation driven by climatological data of the input parameters is compared to cruise measurements, the absolute mean error of 6.9 pmol L$^{-1}$ and the mean error of 3.7 pmol L$^{-1}$ indicate an overall good reproduction of observations (differences between observation and model output were weighted to number of observations in Tab. 5). It has to be noted that on average, the model overestimates OCS concentrations as indicated by the positive mean error, suggesting our emission estimate to be an upper limit to direct oceanic OCS emissions in most regions. Largest deviation from observations are found in the Southern Ocean (vgl. (Staubes and Geogrii, 1993) in Tab. 5), where the model underestimared observations by ~40%. While this can have several reasons, i.e. a possible violation of the underlying assumption of a constant OCS production in regions with deep mixed layers such as the Southern Ocean, or the missing satellite data for CDOM during polar nights, it is a clear indication of the need of more observations from high latitudes. However, this underestimation does not infer with our conclusion drawn for the tropical oceans, where the location of the missing source is derived from top-down approaches."

**We hope that with these modifications and additions, it becomes clearer how our line of arguments leads to the conclusions drawn.**

*Robust conclusions for this study could instead focus on describing the ocean source for the times/locations of the three cruises shown in Figure 1. A qualitative comparison could also be made with previous top-down analysis. This seems to be good ground for an exciting conclusion of consistency between top-down and bottom-up estimates. In this case there appear to be some strong similarities between the bottom -up and top-down estimates. The TranPEGASO cruise covers a section of the Atlantic in Oct/Nov, showing a small source. This is qualitatively consistent with the MIPAS data along the same path in Sept/Oct/Nov. The Oasis cruise covers a small area in the central Indian Ocean in July/Aug showing a sink. This is roughly consistent with a MIPAS Jun/Jul/Aug map and a TES June map that show this region to be on the border between a source and sink.*

**The suggested qualitative comparison of data from a single cruise for OCS is difficult, because the satellite data and atmospheric inversions do not differentiate between direct and indirect emissions. We use our measurements to increase process understanding on a broader scale and use this to address the question of sources and sinks combining direct and indirect sources. As already stated in the text, it is impossible to conclude whether or not the ocean was a net source or sink for direct OCS from TransPEGASO, as only 2 measurements per day were available.**

*ASTRA-OMZ show a strong source in October for the Peru-Chile upwelling region. MIPAS Sept/Oct/Nov do not show this. However, MIPAS is an upper troposphere estimate so it is not expected to provide a close relationship to surface fluxes in regions without strong atmospheric convection such as the Peru-Chile upwelling region. TES provides a lower altitude sensitivity and could provide a better top-down on small regions of sources such as the Peru-Chile upwelling regions. While TES data have only been published for June, TES retrievals for other months are in preparation.*

**As the reviewer correctly mentions, TES would be a better comparison to our combined flux maps, but is unfortunately not yet available. Similarly to our comment above, indirect fluxes contribute significantly to the atmospheric budget and can currently not be differenciated by satellites.**

*Several revisions are needed in the introduction. Page 2 indicates that top-down studies were not consistent with the Kettle bottom-up estimates for sources and sinks. This should be corrected to say that the bottom-up and top-down info does not agree with Kettle. Kettle was a misinterpretation of the bottom-up information from plant studies which was first pointed out in the bottom-up study of Sandoval-Soto et al. and then later confirmed by multiple topdown studies (Campbell et al, Sunthralingam et al, Berry et al, etc.) and other bottom-up studies using chamber (Stimler et al) and canopy (Asaf et al, Maseyk et al) approaches.*

**We thank the reviewer for pointing us to these studies and added/adapted the following lines to the manuscript p. 2, l. 21ff "Nonetheless, current figures for tropospheric OCS sources and sinks carry large uncertainties (Kremser et al., 2016).  While the budget has been previously considered closed (Kettle, 2002), a recent upward revision of the vegetation sink (Sandoval-Soto et al., 2005; Suntharalingam et al., 2008; Berry et al., 2013) led to a gap, i.e. a missing source, in the atmospheric budget of 230-800 Gg S per year (Suntharalingam et al., 2008; Berry et al., 2013; Kuai et al., 2015; Glatthor et al., 2015)(Tab. 1). , with the most recent estimates at the higher end of the range. This revision of vegetation uptake was suggested as to (i) take into account the different deposition velocities of $CO_2$ and OCS within the leaf and base it on GPP instead of net primary production (Sandoval-Soto et al., 2005) as well as (ii) to better reproduce observed seasonality of**

**OCS mixing ratios in several atmospheric models (Berry et al., 2013; Kuai et al., 2015; Glatthor et al., 2015)."**

*The top down evidence from the global scale should be better specified. First it should be clear that there are four independent lines of five independent lines of evidence that point to a tropical source: MIPAS satellite (Glatthor et al), TES satellite (Kuai et al), FTIR (Wang et al, ACP, 2016... this ref isn't in the manuscript but might be added), NOAA and HIPPO observations (Berry et al, Kuai et al, Suntharlingam et al).*

**We agree that we did not fully address all of the mentioned studies. However, only a latitudinal gradient on mixing ratios alone does not point to a tropical hotspot source (i.e. it could also be stronger high-latitude sinks) or an ocean source (i.e. other sources such as anthropogenic sources are possible). We wanted to highlight the magnitude of the missing source suggested by the inverse modelling studies in this section. The fact that highest atmospheric volume mixing ratios are found in the tropical atmosphere does not *per se* contradict our bottom-up emission estimate, as the oceanic emission is still confirmed as one of the dominant sources to the global budget in our study. We thus adjusted the following sentence, including the suggestions from the reviewer p. 2, l. 28: "Based on independent top-down approaches using the MIPAS (Glatthor et al., 2015) and TES (Kuai et al., 2015) satellite observations, FTIR measurements (Wang et al., 2016) as well as NOAA ground based time series stations and the HIPPO aircraft campaign (Berry et al., 2013; Kuai et al., 2015) together with inverse modelling, the missing source of OCS was suggested to originate from the (tropical) ocean."**

*A critical point should be raised to alert the reader to alternative explanations for the top-down trends. In particular, the MIPAS remote sensing data is the upper troposphere (10km) and transport from Asia to the upper troposphere in the deep tropics (e.g. Ashfold et al ACP 2015).*

**We included this point in our manuscript by adding p.2, l. 32: "Other potential sources like e.g. advection from air masses from Asia have been discussed (Glatthor et al., 2015) but not tested.".**

*Recent anthropogenic emission estimates for Asia are not yet sufficient to explain the missing source but they are based on very little bottom-up data from Asia (Campbell et al 2015). Other hypotheses could be mentioned as well such as a soil source which has been shown in a recent survey of global soils but not particularly large in the tropics (Whelan et al ACP 2016). Biomass burning is another but the most recent review of emission factors shows a relatively small source (Campbell et al 2015).*

**We added a sentence on the potential of biomass burning as the missing source p.14, l.10: "While biomass burning is known to emit OCS and is present close to the assumed source region, e.g. around Indonesia, the most recent review of emission factors result in a source too small to close the atmospheric budget (Campbell et al., 2015)". Two other points are already described in our conclusion section on p. 14, line 12ff (other anthropogenic sources, now extended "However, Lee and Brimblecombe (2016) reevaluated the anthropogenic emissions of OCS and its precursors and provide a higher number than previously considered of 598 Gg S yr$^{-1}$. They attribute the largest direct OCS emissions to biomass and biofuel burning as well as pulp and paper factory, and the largest CS$_2$ emissions to the rayon industry. Hence, a hot spot of anthropogenic emissions in the Asian continent might be a potential candidate, together with atmospheric transport, to produce atmospheric mixing ratios as observed by the satellite.") and p.14, l.18ff (soils).**

*The Van Hobe study was included but more could be done to explain what other cruise data is available. The introduction needs to explain how the cruise measurements and ocean box modeling fit within the context of previous cruise measurement and ocean box modeling. Were these cruises in seasons or locations that have others have not gone?*

**We agree that an overview on previous cruise data should be stressed more in this manuscript, which we do with the following addition apart from the new table 5. and box model description in section 2.4 described in our first comment above.**
**We add a more detailed description on previous OCS, CS2 and DMS measurements in the surface ocean in the introduction, p.3, l. 3ff: "OCS and its atmospheric precursors are naturally produced in the ocean. In the surface open ocean, OCS is present in the lower picomolar range <100 pmol L$^{-1}$, and has been measured on numerous cruises to the Atlantic (Ulshöfer et al., 1995; Ulshöfer et al., 1996; Ulshöfer and Andreae, 1998; Von Hobe et al., 1999) (including 3 latitudinal transects (Kettle et al., 2001; Xu et al., 2001), the Indian Ocean (Mihalopoulos et al., 1992), the Pacific Ocean (Weiss et al., 1995) and the Southern Ocean (Staubes and Georgii, 1993). OCS is produced photochemically from chromophoric dissolved organic matter (CDOM) (Andreae and Ferek, 2002; Ferek and Andreae, 1984) and by a not fully understood light independent production pathway that depends on temperature and CDOM concentration (Flöck et al., 1997; Von Hobe et al., 2001) Dissolved OCS is efficiently hydrolyzed to $CO_2$ and $H_2S$ at a rate depending on pH and temperature (Elliott et al., 1989). $CS_2$ has been measured in the Pacific and Atlantic oceans in a range of 7.2-27.5 pmol L$^{-1}$ (Xie et al., 1998) and during two Atlantic transects (summer and winter) in a range of 4-40 pmol L$^{-1}$ (Xu et al., 2001). It is produced photochemically (Xie et al., 1998) and biologically (Xie and Moore, 1999), and no significant loss process other than air-sea gas exchange has been identified (Xie et al., 1998). DMS is present in the lower nanomolar range in the surface ocean and has been extensively studies in several campaigns, summarized in a climatology by Lana et al. (2011). DMS is biogenically produced and consumed in the surface ocean, as well as photo-oxidized and ventilated by air-sea exchange (Stefels et al., 2007)."**

*The introduction or discussion could also compare the modeling approach here to what has been done previously. In particular the recent paper by Launois et al.*

**We now discuss the comparison to Launois et al., 2015 on p. 7, l. 4ff: "Launois et al. (2015) to implemented parameterizations for light-independent production, hydrolysis and air-sea exchange similar to vH2003 in the 3D global ocean model NEMO-PISCES. The main differences to the approach used here is the lack of accounting for mixing in our model (discussed in section 3.2.2 (which will theoretically lead to higher simulated concentrations in our case) and the application of a photoproduction rate constant in our model that incorporates information from three open ocean basins in contrast to one from a study in the North Sea (Launois et al., 2015)."**
**A section on the development of the box model used in this study is now added on p. 6., l. 23ff.**

*Some comments may be helpful on alternative approaches for validating these flux estimates. Spatial gradients in atmospheric mixing ratios have been used recently (Berkelhammer et al below). Are other approaches also possible? (M. Berkelhammer, H.C. Steen-Larsen, A. Cosgrove, A. Peters, R. Johnson, M. Hayden and S.A. Montzka (Minor Revisions, July 2016) Radiation and atmospheric circulation controls on carbonyl sulfide concentrations in the marine boundary layer. Journal of Geophysical Research (available upon request).*

**Validation of the flux estimates would need different methods for different scales. Using atmospheric gradients could help to qualitatively validate sources and sinks, but since OCS is such a long-lived gas, the volume mixing ratio of OCS on larger scales is determined by many factors among which direct emissions are only one part (i.e. indirect emissions, conversion in the atmosphere, boundary layer height, trajectory history,…). The mentioned study shows that the ocean can be a source or have a zero net flux regionally, which qualitatively confirms our results, but of course cannot quantitatively validate global emission estimates.**
**Fluxes are physically defined by F=k x Δc, and computing them by measuring the concentration gradient is an established method that has been validated before (Johnson, 2010). Quantitatively validating the emission estimate on the local scale would mean using an independent way of measuring OCS emissions. This can be done by direct flux measurements, e.g. eddy covariance. As**

OCS is such a long-lived gas, any validation on the global scale needs the global consideration of all sources and sinks, and atmospheric inversions can be used to establish a budget, but need the bottom-up validation of measurements themselves. Any additional data constraints from e.g. time series stations in the tropics and more measurements at sea, tested against the box model, would be beneficial in that respect.

*Section 2.3 should describe how the box model relates to the measurements. This is done in the results section "Following an earlier study (von Hobe et al., 2003), we use our observations ..." but belongs in the methods. A few additional sentences of explanation may be helpful.*

We shifted the mentioned part to the method section. Additionally, we clarified that the box model simulations of the two cruises were used as case studies to derive the photoproduction rate constant and validate against data from the tropical region, as a proof-of-concept for the global application of the model on p. 8, l. 15ff. "To extend the p-CDOM-relationship for other ocean basins, we use the two cruises OASIS and ASTRA-OMZ as case studies for parameter optimization of the photoproduction rate constant p. The photoproduction constant p in the case study simulations was fitted individually for periods of daylight >100 W m$^{-2}$ (Fig. 2, blue lines) with a Levenberg-Marquart optimization routine in MatLab version 2015a (8.5.0), by minimizing residuals between simulated and hourly averaged measurements. Different starting values were tested to reduce the risk of the fitted p being a local minimum."

*Why was the parameter p chosen for fitting the model as opposed to the numerous other parameters. Were other parameters also examined? If not then perhaps this should be stated as an important next step for future work.*

The parameter p was chosen for fitting since this is the one that is the most difficult to determine from measurements when a wavelength-integrated approach is chosen as is done in our model. We added on p.8 l.10ff: "The rate coefficients for hydrolysis, light-independent production and air-sea exchange are all reasonably well constrained and parameterizations have been derived from dedicated laboratory and field experiments (hydrolysis, air-sea exchange) or from nighttime OCS observations in several regions assuming steady-state (dark production, (Von Hobe et al., 2001)). On the contrary, the photoproduction rate constant p is not well constrained and no generally applicable parameterization exists. von Hobe (2003)have made a start of parameterizing p in terms of CDOM absorption, and found this to be dependent on the exact model setup used with respect to wavelength integration and mixed layer treatment."

*Why was the von Hobe et al., 2003 study used but not other studies? What is the spatial and temporal extent of the Von Hobe data?*

The model from von Hobe et al., 2003, is the most recent version of the box model which was further developed in our approach. We added a paragraph on the evolution of this model (beginning of section 2.4, p. 6), see also our comment 2, point (1) above.

*"global radiation I was" not sure what "I" is*

The "*I*" should have been in italics, as it is the symbol for global radiation in the equations, which we now corrected.

*page 6, explain what you mean by "case study simulations"*

We meant our two cruises as case studies for the global model, which we clarify by adding p. 8, l.15: "To extend the p-CDOM-relationship for other ocean basins, we use the two cruises OASIS and ASTRA-OMZ as case studies for parameter optimization of the photoproduction rate constant

**p. The photoproduction constant p in the case study simulations was fitted individually for periods of daylight >100 W m$^{-2}$ (Fig. 2, blue lines) with a Levenberg-Marquart optimization routine in MatLab version 2015a (8.5.0), by minimizing residuals between simulated and hourly averaged measurements."**

*define "CTD profiles"*

**now defined "…CTD (conductivity, temperature, depth)…" p.8, l.1**

*The methods section should also include a summary of the time and location of the 3 cruises.*

**We added a new section 2.1 to summarize the study sites and cruises.**
**p.4, l. 5: "Several cruises were conducted to measure the trace gases OCS (OASIS, TransPEGASO, ASTRA-OMZ) and CS$_2$ (TransPEGASO, ASTRA-OMZ). Cruise tracks are depicted in Fig. 1. The OASIS cruise onboard RV SONNE I to the Indian Ocean started from Port Louis, Mauritius to Male, Maledives in July and August 2014, where mainly oligotrophic waters were encountered. TransPEGASO was an Atlantic transect starting in Gibraltar leading to Buenos Aires, Argentinia and Punto Arenas, Chile. It took place in October and November 2014 and covered a variety of biogeochemical regimes. ASTRA-OMZ onboard RV SONNE II started in Guayaquil, Ecuador and ended in Antofogasta, Chile, in October 2015. Although 2015 was an El Nino year, upwelling together with high biological production was still encountered during the cruise (Stramma et al., 2016)."**

*The section "2.1 Measurement set-up for trace gases" present a different method for each cruise. It would be helpful if this section also summarized the impact of having different methods on the different cruises in terms of different precision and other factors that may or may not influence the quality of these measurements.*

**The different methods are discussed in section 2.1, with detailed description of the different methods and precisions for all methods (i.e. OA-ICOS for OCS during OASIS/ASTRA-OMZ, GC-MS for OCS during TransPEGASO as well as for CS$_2$ during TransPEGASO and ASTRA-OMZ) including standards and calibration procedures, temporal resolution of the measurements, precision etc. We added the following on p. 5, l. 29.: "The systems are calibrated against a standard each, but had not been directly intercompared. Still, our measurements are consistent with previous measurements using independent methods as discussed in section 3.2.1. and 3.3".**

*Table 3 missing km^-2 in TransPEGASO flux*

**Now corrected**

*"an non-negligible" to "a"*

**Now corrected**

*Some description is needed of the error associated with assuming a constant atmospheric mixing ratio on TransPEGASO. Seasonal and spatial variation in atmospheric mixing ratios can be on the order of 100 ppt.*

**We performed a sensitivity test for a scenario with 450 and 550 ppt, and added the following sentences to the manuscript p. 6, l. 6: "As air volume mixing ratios of OCS vary over the course of a year, we performed a sensitivity test for a scenrio of 450 and 550 ppt and found mean deviations of +7.8 and -7.8 % respectively."**

*"leaving the missing source still explained" should be "unexplained"?*

**Now corrected**

*Again this is an important contribution of new, high quality data and a well written manuscript. The authors present a compelling approach and with further data could provide a key to closing the global COS budget.*

Kettle, A. J.: Extrapolations of the flux of dimethylsulfide, carbon monooxide, carbonyl sulfide and carbon disulfide from the oceans, PhD, Graduate Program in Chemistry, North York, Ontario, 2000.

[revised manuscript text omitted]

---

## Author Comment (AC2) · 22 Nov 2016

We thank the reviewer#2 for the review that helps to improve our manuscript. In the following, we address the raised points directly, with the *review in italics* and **our reply in bold** font.

*I find it striking that such broad generalizations are made from data that cover a fairly small portion of the global ocean. The authors have a great deal of new observations to address the important issue of quantifying OCS fluxes from the ocean. But to draw broad conclusions without considering the previous data more carefully is inappropriate. Their data add to the picture in useful and interesting ways. To "conclusively answer the question of whether the missing OCS source... can really be ascribed to the direct OCS emissions from tropical oceans" would seem to require another level of effort that isn't yet part of this manuscript.*

**We agree that we need to add information on a comparison of our model simulation with previous cruises, which we now added in a new Table (Table 5) where we compare previous studies from various seasons and regions with our box model simulation (>4000 measurements). We discuss this in a new section "3.2.1 Comparison to previous ship-based measurements" (p.11). We find overall good agreement even though we compare monthly climatological means from the box model simulation with cruise data. This further increases the confidence in our model simulation.**
**In addition, we emphasize again that the box model had already been established previously and tested with data from several cruises and seasons (p. 6, l.22ff). We further improve the model by consolidating the aCDOM dependence of the photoproduction rate constant *p* and constrain the model by important new data from tropical regions which had been missing previously.**

**See also our response to a similar comment by reviewer #1.**

**New Tab. 5**

[revised manuscript text omitted]

*I realize that the model helps the authors extrapolate their results to broader scales, but the results derived are only as good as the data considered by the model. If the authors really hope to be able to "constrain the variability of OCS emissions in the tropic[al ocean]" then I would think they would have to consider the details of previous ocean-going measurements (ocean basin, ocean regime, season, etc.) together with their new data to determine if, in fact, that most of the major global ocean regimes have been adequately sampled to allow such a conclusion.*

**Additional to considering previous measurements (see above), we also added a comparison of the water properties of our cruises and the assumed source region (Pacific Warm Pool, new Tab. 4). The properties encountered during our 2 cruises which we use to further improve and validate the box model (OASIS, ASTRA-OMZ) comprise the range of properties found in the Pacific Warm Pool except for SST and windspeed. We discuss this in p. 10, l. 20ff in the revised manuscript.**

**New Tab. 4:**

| Parameter | OASIS | ASTRA-OMZ | Pacific Warm Pool |
|---|---|---|---|
| SST [°C] | 27.0±1.0 | 19.6±2.6 | 28.9±0.9 |
| SSS [g kg$^{-1}$] | 35.0±0.3 | 35.1±0.3 | 34.5±0.42 |
| wind speed [m s$^{-1}$] | 8.2±1.7 | 7.5±1.8 | 5.3±0.4 |
| a$_{350}$ [m$^{-1}$] | 0.039±0.02 | 0.146±0.02 | 0.050±0.08 |
| I [W m$^{-2}$] | 226.5±303.0 | 196.4±283.1 | 206.4±286.6* |
| SR [J m$^{-2}$] | 1.9 10$^7$±1.7 10$^6$ | 1.6 10$^7$±4.5 10$^6$ | 8.9 10$^6$±1.3 10$^6$ |
| pH [-] | 8.03±0.01 | -** | 8.07±0.01 |
| MLD [m] | 43.3±15.8 | 18.9±7.5 | 35.9±14.1 |

**Discussion of Tab. 4 in the text:**
**p.10, l.20: "The water masses encountered during the cruises to the Indian Ocean (OASIS) and eastern Pacific (ASTRA-OMZ), which are used to constrain the global box model, differ considerably with respect to the properties relevant for OCS cycling and thus span a large range of possible OCS variability. The properties encountered during these two cruises comprise or exceed the ones of the Pacific warm pool (climatological averages, Tab. 4), which is where the missing source has been located (Glatthor et al., 2015; Kuai et al., 2015). Both higher SST and lower wind speeds (Tab. 5) would decrease the OCS sea surface concentrations in the ocean and thus decrease emissions to the atmosphere: higher SSTs favor a stronger degradation by hydrolysis (Elliott et al., 1989), and lower wind speeds decrease the transfer velocity k and thus the emissions to the atmosphere. Lower integrated daily radiation (SR in Tab. 4) in the Pacific warm pool also points to lower OCS production. Hence, our new OCS observations presented here likely span the range of emission variability in the tropics."**

**Note that the box model has been used in a number of studies comparing observed and simulated OCS for individual cruises from different parts of the oceans and always yielded reasonable results.**

*For example, it is informative and important that their results are consistent with an upper flux limit from Kettle et al., 2002, but no mention is made of how that consistency actually increases our understanding of total OCS flux from the ocean. If the sampled regions were similar, then this is a confirmation of that original estimate, but potentially not much progress in understanding the broader-scale contribution of the ocean to atmospheric OCS abundances.*

**Kettle et al. 2002 built his global model on data from a transect through various regimes in the Atlantic, so our new data from the Indian, Atlantic (comparable) and East Pacific increase the data bases and model test cases for two new ocean basins (discussed on p.6 l.32ff in the revised manuscript).**
**"(Kettle, 2000; Kettle, 2002) applied a similar version of vH2003 globally, which included an optimized photoproduction constant from Atlantic transect cruise data, an optimized constant light-independant**

**production and a linear regression to obtain CDOM from chlorophyll a. In comparison to K2000, we use (i) a new way of determining the photoproduction rate constant incorporating information from three ocean basins, (ii) the most recent parameterization of light-independent production available, and (iii) satellite observations for sea surface CDOM instead of an empiric relationship based on chlorophyll a."**

*Suggestion: scale back the broad conclusions and focus on your results and how they compare with others, or consider the broader literature on OCS flux measurements (observation-based and model-derived) in a more detailed fashion to determine if the available data allow an accurate quantification of the global and, most importantly, tropical OCS flux from the ocean.*

**As described above, we considered the broader literature by (i) comparing our model simulations to 9 studies and (ii) describing more detailed in which regions the box model had been tested previously**. **With changing the title to "Direct Oceanic OCS emissions are unlikely to account for the missing source of atmospheric carbonyl sulfide" we acknowledge the higher uncertainty associated with the indirect emission estimates.**

*On uncertainties. Any comparison of derived oceanic fluxes with a shortfall needs to fully consider uncertainties. Many uncertainties are discussed (air-sea exchange, mixed layer depth, parameterization of production, etc.,) but aren't explicitly included in the derivation of the direct flux of 130 GgS/yr and in the discussion of the global budget discrepancy (no uncertainty is provided on this number).*

**We restructured the results part and discuss uncertainties now in a new section "3.2.2 Uncertainties" (p. 12, l. 1ff). From our comparison to measurements from section 3.2.1, we use the mean error to derive an uncertainty for the global emission estimate, which results in an uncertainty of 50 Gg S per year.**

**p. 12, l.1: "Simulated concentrations and fluxes carry uncertainties from input parameters and process parameterizations. One major uncertainty associated with the mixed-layer box model approach arises from the fact that it does not adequately account for downward mixing and vertical concentration gradients within the mixed layer. Under most circumstances, and especially in the tropical open ocean, where hydrolysis greatly exceeds surface outgassing and low $a_{350}$ makes photoproduction extend further down in the water column, the model tends to overestimate the real OCS concentrations, as was shown for our two cruises above. Therefore, we deem the fluxes from our global simulation to represent an upper limit of the true fluxes. Only at high latitudes would we expect more complex uncertainties, because hydrolysis at low temperatures is slow and only photoproduction and loss by outgassing are directly competing at the very surface. Other uncertainties are associated with the calculation of the photoproduction rate. The wavelength of 443nm combines the absorption of detritus and CDOM, which could have an impact especially in river plumes, where terrestrial material is advected into the ocean. As it is the CDOM that is important for photochemistry, assuming the 443nm is purely CDOM would lead to an overestimation of photoproduction and thus is a conservative estimate. It also has to be noted that a single spectral slope from 443nm to 350nm in the global simulation is a simplification. Furthermore, using a wavelength integrated photoproduction rate constant instead of a wavelength resolved approach, which would take global variations in the CDOM and light spectra into account, is an additional simplification. It has been shown that this does not lead to large differences regionally (von Hobe, 2003), but potentially could lead to variations globally. Our p-CDOM-relationship is a first step for constraining this variability globally in one parameterization, as it incorporates optimized photoproduction rate constants optimized to observations and thus accounting for differences in the light and CDOM spectra. More data from different regions can help to further constrain this relationship in future studies. However, despite these simplifications, the simulated concentrations agree very well with previous observations (n>4000, Tab. 5).**

**To test the sensitivity of our box model to the photoproduction rate constant, we performed a sensitivity test with a photoproduction increased by a factor of 5 in the tropical region 30°N-30°S, note that this factor is considerably larger than the uncertainty in the p-CDOM-relationship). This leads to an annual mean concentration of 35.1 pmol L$^{-1}$ in the tropics (30°N-30°S), resulting in tropical direct emissions of 160 Gg S as OCS per year. The efficient hydrolysis in warm tropical waters prevents OCS concentrations from accumulating despite the high photoproduction, and still results in emissions too low to account for the missing source.**
**With an mean error of 3.7 pmol L$^{-1}$ in the OCS surface water concentrations added to (subtracted from) the modelled concentration and subsequent calculation of fluxes using annual mean climatologies for wind, pressure and SST (same data sources as global simulation forcing data), we calculate an uncertainty of 60%, which translates into a total uncertainty of the integrated global flux of 80 Gg S yr$^{-1}$."**

*Similarly, the origin of stated uncertainties in the derivation of indirect fluxes from DMS are not discussed, but I would imagine are substantially larger then estimated. Without a fair treatment of these uncertainties, it isn't possible to gauge the true magnitude of the budget shortfall, which is a main point of the manuscript.*

**We agree that the uncertainty of the DMS conversion is large, which is why we took the highest published estimate of 0.7% OCS yield from DMS oxidation as a conservative estimate for remote, pristine atmospheric conditions. The emission pattern of DMS does not point to a specific tropical source. This implies that if emissions and subsequent oxidation of DMS was to account for the missing source, a conversion process specifically favored in the tropical troposphere is required. We now discuss the conversion factor of OCS to DMS in greater detail (p. 12, l. 31ff). Any additional investigation on the conversion factor is beyond the scope of this paper, as the spatial pattern of the missing source would then require an atmospheric, not an oceanic driver to produce a pattern with a hot spot of OCS production in the Pacific warm pool.**

**The revised section on OCS emission factors p. 12, l. 31:**
**"A significant contribution to the OCS budget in the atmosphere results from oceanic emissions of DMS and CS$_2$ that are partially converted to OCS on time scales of hours to days (Chin and Davis, 1993; Watts, 2000; Kettle, 2002). A yield of 0.7 % for OCS is used for the reaction of DMS with OH (Barnes et al., 1994), which results in a global oceanic source of DMS from OCS of 80 (65 - 110) Gg S yr$^{-1}$ based on the procedure discribed in section 2.5. The uncertainty range of 65-110 Gg S yr$^{-1}$ originated from the uncertainty in oceanic emissions, not the conversion factor. This conversion factor is much more uncertain, as the formation of OCS from DMS involves a complex multi-step reaction mechanism that is not fully understood. It has been shown in laboratory experiments, that the presence of NO$_x$ reduces the OCS yield considerably (Arsene et al., 2001), which would make our indirect emission estimate an upper limit. However, the yield was measured under laboratory conditions, and may be different and more variable under natural conditions. \\**
**DMS emissions do not show a pronounced hot spot in the Pacific warm pool region, but as DMS transports much more sulfur across the air-sea interface than OCS, even low changes in the OCS yield could affect the atmospheric budget of OCS. As the spatial oceanic emission pattern of DMS does not reflect the spatial pattern of the assumed missing source, a locally specific tropospheric change in the conversion yield would be one potential way of bringing the patterns in agreement. While it is possible that the OCS yield could vary under certain conditions, e.g. it cannot be excluded that the low OH concentrations in the broader Pacific warm pool area as suggested by Rex et al. (2014) influence the yield, the (local) increase of the conversion factor would need to be in the order of a factor of 10-100."**

*Details: P1, line 7-8. It only has implications for GPP derived from OCS observations on certain scales, not all.*

**Now changed**

*p.2, line 25-26. It also makes much more chemical sense given our understanding of how COS interacts with carbonic anhydrase in leaf waters.*

**We added this point on p. 2, l. 25ff: "This revision of vegetation uptake was suggested to (i) take into account the different deposition velocities of $CO_2$ and OCS within the leaf and base it on GPP instead of net primary production (Sandoval-Soto et al., 2005) as well as (ii) to better reproduce observed seasonality of OCS mixing ratios in several atmospheric models (Berry et al., 2013; Suntharalingam et al., 2008; Glatthor et al., 2015; Kuai et al., 2015)."**

*Table 1 and p. 9 line 27-30. The paper from Suntharalingam does not state or suggest that the missing source is oceanic.*

**This is correct, so we took the citation out of the text on p. 9. We still leave it in Tab. 1, as it is one of the missing source estimates. The caption of Tab. 1 specifies that if the missing source is assigned to the tropical oceans (which other studies do), then this would result in the total ocean flux listed.**

p.3 line 13. Underrepresentation of global flux is also true for the measurements made here, despite the use of a box model for extrapolation.

**This is true theoretically, but the difference in photoproduction constants between the open Atlantic ocean and the North Sea has been compared specifically in Uher and Andreae (1997a). We thus adjusted the sentence to: p. 3, l. 25: "However, the oceanic OCS photoproduction in the ocean model included a parameterization for OCS photoproduction derived from an experiment in the North Sea (Uher and Andreae, 1997b), which might not be representative for the global ocean, as indicated by an order of magnitude lower photoproduction constants in the Atlantic ocean compared to the German Bight (Uher and Andreae, 1997a)."**

*Figure 3, and line 10, p. 9. Is R = 0.7 or R^2 = 0.7 in this relationship? These different values seem given for one relationship. Also, given the non-normal distribution of these results this value is not as significant as one might presume.*

**Because the data is not normally distributed, we used Spearman's rank correlation coefficient r, which is correctly displayed in the figure, and now also corrected in the text. By relating the photoproduction rate constant to a350, we establish a semi-empirical relationship, which is a reasonable and globally available, but not perfect proxy (i.e. CDOM is only an approximation for the precursor).**

*The conclusion section is a bit unusual in that it includes speculations not supported by the work. It also reads like a research planning document. On what basis do the authors conclude (p. 11, line 22) that observations could be reproduced "without increasing the vegetation sink"? An extensive body of recent literature has suggested that the interaction between OCS and vegetation is best explained by a substantially larger sink than discussed in earlier papers; to discount those studies without substantial evidence is inappropriate, making this conclusion one that does not follow from the evidence presented in the paper.*

**In the last chapter we meant to conclude our results and give a systematic outlook on further research. We changed the section title to "Conclusion and Outlook". Nevertheless, we still want to discuss possible ways of reducing uncertainties in the emission estimates and closing the budget. Suggestions that could be misinterpreted as discounting previous research have been deleted.**

**References**

[revised manuscript text omitted]

---

## Author Comment (AC3) · 22 Nov 2016

We thank S. Belviso for his comment to our manuscript, and will address the points raised in the following, with *his comments in italics* and **our replies in bold** font. We are pleased to discuss any further emerging questions.

*The bottom-up approach of COS concentrations in the oceanic mixed layer used in this study relies on the relationship shown in Fig. 3 where the photoproduction rate constant (or quantum yield) at 350 nm (p350) is linearly correlated to the CDOM absorption coefficient at 350 nm. A similar study was carried out by von Hobe et al. (2003) but, in that study, the authors investigated p313 and a313. A wavelength of 313 nm was used because "it falls into the center of the wavelength band for UV and coincides well with the wavelength where COS surface production rate spectra show a maximum in tropical waters (Weiss et al., 1995)." I wonder why the authors calculated a new relationship at 350 nm which is the wavelength where COS surface production rate spectra show values about twice lower than at 313 nm (see Fig. 2C of Weiss et al. (1995)). Can you clarify this point?*

**It is true that the quantum yield for OCS is largest at smaller wavelengths, as found e.g. in incubation studies by Weiss et al. (1995) with seawater samples from the Pacific Ocean. We chose to use the a350 here, as the parameterization for dark production is also based on a350 and thus can rely on the same input parameter. That this assumption is valid is shown in our Fig. 3, as the linear correlation for *p* related to a350 yields a similar shape as the p-a313 relationship in von Hobe (2003) Fig. 5. The crucial point however is that *p* is fitted in a consistent way, as *p* is wavelength integrated and is thus only valid for the same light wavelengths in the forward and in the inverse mode. As we use the same model as von Hobe (2003) with respect to the UV light field, we can use his fitted *p* and relate them to a350 together with our new values.**

*The average action spectrum of COS used by von Hobe et al. (2003) is the one established by Weiss et al. (1995) from incubation experiments carried out in tropical and Antarctic regions. Weiss et al. compared the average action spectrum from their work to that of Zepp and Andreae (1990) who investigated coastal North Sea water samples.*
*Weiss et al. noted that differences in DOM origin between the two works may explain the pronounced discrepancy in relative action spectra. My interpretation of data shown in Fig. 3 is that the quantum yield of COS is about twice higher in the surface waters of the Indian and Pacific oceans than in the Atlantic. Because the COS quantum yield exhibits quite large variations even far from coastal areas, I wonder how the Lennartz et al.'s relationship can be representative for the global ocean.*

**Your interpretation to attribute the scatter in Fig. 3 to variations in the apparent quantum yield (or the concentration of precursors in relation to CDOM) is correct and certainly exhibits variability globally. A box model like the one used here is always a simplification, and the photoproduction rate constant is certainly the part where the model can be better constrained when more data is available. However, it has to be noted that this linear relationship as presented in Fig. 3 contains information from 3 very different oceanic regions with respect to the biogeochemistry, which is the reason for the scatter, as it takes into account the differences in spectral shape for light and CDOM absorbance. Given the fact that this one linear relationship does not only reproduce the data from the two cruises that cover a low CDOM (OASIS) and a high CDOM (ASTRA-OMZ) area, but also simulated global concentrations that agree well with observations (see below, new Tab. 5 in revised manuscript), it can be used as a first approximation for a global relationship. A more exact identification of the precursor instead of approximating it by CDOM would reduce the scatter. However, this relationship will not be the crucial point when addressing the missing source question, as assuming a higher quantum yield as in the Atlantic will lead to an overestimation of concentration and thus represents a conservative estimate in our case.**

The latitude-time plots in Fig. S5 (this work) and in Fig. 8 of Launois et al. (2015) look pretty much the same in terms of hydrolysis rates in tropical regions (1 to 2 pmol L-1 h-1). How is this possible knowing that your model predicts considerably lower COS levels in tropical waters?

**This difference is indeed present, and we will describe our way of calculation here using a concrete example: For the example, we assume a mean SST of 25°C, a salinity of 35 psu and a pH of 8.1. According to the implementation of the rate constant by Elliott et al. (1989) taken from von Hobe (2003), this results in a rate constant of $3.7 \times 10^{-5}$ $s^{-1}$, a range which seems reasonable given Tab. 1 in Elliott et al. (1989). The hydrolysis rate is then obtained by multiplying this rate constant with the actual concentration assuming first order kinetics. This would lead to the following hydrolysis rates when assuming a concentration of 10 pmol $L^{-1}$ as found in the simulation discussed in the Lennartz et al. manuscript, or concentrations taken from S-Fig. 1 in Launois et al. (2015) (by eye) of 100 pmol $L^{-1}$:**

| Concentration | Hydrolysis rate |
|---|---|
| pmol $L^{-1}$ | pmol $L^{-1}$ $hr^{-1}$ |
| 10 | 1.33 |
| 100 | 13.3 |

**The 1.33 pmol $L^{-1}$ $hr^{-1}$ in relation to 10 pmol $L^{-1}$ of OCS agree well with the values calculated for the tropics in the Fig. 5 in the supplement of the Lennartz et al. manuscript.**

My last concern, to conclude, relates to the lack of validation of your extrapolations from published inventories. I refer to that of Mihalopoulos et al. (1992) who gathered cruise observations of the supersaturation ratio (SR) of COS in coastal areas and in the open ocean (their Tables 1 & 2) and of the latitudinal variations of seawater COS concentrations (their Fig. 2). The calculation of oceanic fluxes requires 6 variables (kw,T, wind speed, Cw, Ca and H). That of SR only requires 4 variables (Cw, Ca, T and H). The direction of the flux can be assessed from SR values (SR<1 = sink; SR>1 =source). I think you should try to evaluate simulated monthly maps of SR against the inventories of Mihalopoulos et al. (1992). Scanned data plots can be digitized using online facilities.

**This is certainly a point we will address in the revised version of the manuscript. We do not only include comparisons to observations from the study you suggest, but add 8 more studies to compare our model output against, including measurements from several Atlantic transects, the Indian and the Southern Ocean (>4000 measurements). We find an overall good agreement, even when comparing to our model output driven by climatological means (absolute mean error 6.9 pmol $L^{-1}$, mean error 3.7 pmol $L^{-1}$). As a direct comparison, we use mean concentration values presented in the papers, and compare them to corresponding months and regions in our model output, still acknowledging that the climatological input parameters of the model might not fully represent the conditions encountered during the OCEAT II and III cruises. Using the mean values provided in section 3.2 (b, open ocean) , we obtain the mean value 19.9 pmol $L^{-1}$ during the two cruises opposite to 11.2 resp. 17.7 pmol $L^{-1}$ in our climatological means. The minor discrepancy may be explained by the fact that 20 resp. 14 measurements of unknown time of the day are compared to climatological means.**

**References**

Elliott, S., Lu, E., and Rowland, F. S.: Rates and mechanisms for the hydrolysis of carbonyl sulfide in natural waters, Environmental Science & Technology, 23, 458-461, 10.1021/es00181a011, 1989.

Launois, T., Belviso, S., Bopp, L., Fichot, C. G., and Peylin, P.: A new model for the global biogeochemical cycle of carbonyl sulfide - part 1: Assessment of direct marine emissions with an

oceanic general circulation and biogeochemistry model, Atmos. Chem. Phys., 15, 2295-2312, 10.5194/acp-15-2295-2015, 2015.

von Hobe, M.: Photochemical and physical modeling of carbonyl sulfide in the ocean, Journal of Geophysical Research, 108, 10.1029/2000jc000712, 2003.

Weiss, P. S., Andrews, S. S., Johnson, J. E., and Zafiriou, O. C.: Photoproduction of carbonyl sulfide in south pacific ocean waters as a function of irradiation wavelength, Geophysical Research Letters, 22, 215-218, 1995.

---

## Author Comment (AC4) · 22 Nov 2016

**Dear Cedric,**
**Thank you for your detailed comment to our manuscript. I will address your points in the following and am pleased to answer any remaining questions:**

*Dear Sinikka,*
*I read your paper with great interest and I have a few comments that relate primarily to the photoproduction part of your work. Note I generally support the reviewers' comments on better acknowledging and including uncertainties associated with your estimates. This is especially important because the paper's conclusions contradict the main findings of other published work.*

**Thank you for bringing up the point of better acknowledging the uncertainty in the photoproduction again, which has been addressed in the reply of the reviewers in the revised manuscript. While we contradict the conclusion of Launois et al. (2015), i.e. that direct emissions could account for the missing source, we do not contradict the many oceanic measurements that had been performed in various open ocean regions in different ocean basins and across different seasons. We hope that this point is clarified in the revised manuscript where we added a table (Tab. 5) to compare to >4000 previous measurements.**

*In particular:*
*1. The photoproduction rate calculations deserve to be explained in more details, especially when it comes to the global calculations. In particular, are the rates reported integrated over the mixed layer (as is suggested by equation 2) or are they an average for the mixed layer, or at a discrete depth (as would be suggested by the pmol L-1 h-1units reported on figure S5)? The caption mentions production rates but that's it. If the rates are integrated over the mixed layer then the units of photoproduction rates should be in units of pmol per unit time and unit AREA (not per unit VOLUME). It would be good to clarify this because it would help facilitate comparisons with other approaches.*
*Note I attached (see supplement) a number of figures of calculated rates using the Fichot and Miller (2010) model implemented with the single wavelength-resolved apparent quantum yield of Weiss et al. (1995) for comparison purposes and to illustrate the differences in photochemical rates calculated at different depths or for different depth ranges (mixed layer, and sunlit layer).*

**Thank you very much for providing the plots to illustrate the point of the differences in photochemical rates for different depths. The model in our study treats the mixed layer as one single box and integrates the photoproduction rate constant over this respective depth, as described in equation 2. Thanks for pointing us to the mistake in the caption, the photoproduction rates illustrated in S-Fig. 5 is the average for the mixed layer, which will be corrected in the revised manuscript.**

*2. The attempt to constrain the variability in the photochemical rate constant p is welcomed, but it should also be recognized in the paper that using wavelength-independent p for the global calculations can also lead to significant and potentially large uncertainties in the calculated rate. In the von Hobe (2013) paper, it was suggested this had a little impact, but I believe the comparison using wavelength-independent and wavelength-resolved constants was made at the same location where there should be minor differences in the spectral characteristics of the UV downwelling irradiance spectrum. For global calculations, this simplification can be much more problematic because the spectral characteristics of the downwelling irradiance can vary quite dramatically with latitude or with atmospheric conditions. This is potentially a large source of uncertainty that would be good to discuss in the paper.*

**We will highlight the difference between using a wavelength integrated p and a wavelength resolved p in the revised manuscript in the section about uncertainties. A dependence of the photoproduction rate constant on CDOM makes sense from a chemical point of view as described in von Hobe et al. (2003), but the exact shape of this relationship remained to be defined. Our**

attempt to constrain a global relationship includes photoproduction constants that account for these differences as they are optimized in an inverse set-up of the model. With data from different oceanic regions across three basins, we suggest this relationship as a first step to account for this variability. As is shown by our comparison to measurements from previous cruises (new table 5), this simplification simulates OCS concentrations in a range similar to previous measurements despite the uncertainties mentioned.

To test the sensitivity for the photoproduction rate constant in relation to the missing source, we performed a simulation with an arbitrarily increased photoproduction by a factor of 5 in the tropics. This factor is much larger than the uncertainty in our p-CDOM-relationship. This simulation yields tropical (30°N-30°S) concentrations of 35.1 pmol/L and an integrated flux of 160 Gg S yr$^{-1}$ from this area. The strong hydrolysis in warm tropical waters prevents the OCS from accumulating and results in tropical emissions too low to account for the missing source.

We added the following paragraph to the revised manuscript, p. 12, l. 21ff.:
"To test the sensitivity of our box model to the photoproduction rate constant, we performed a sensitivity test with a photoproduction increased by a factor of 5 in the tropical region 30°N-30°S, note that this factor is considerably larger than the uncertainty in the p-CDOM-relationship). This leads to an annual mean concentration of 35.1 pmol L$^{-1}$ in the tropics (30°N-30°S), resulting in tropical direct emissions of 160 Gg S as OCS per year. The efficient hydrolysis in warm tropical waters prevents OCS concentrations from accumulating despite the high photoproduction, and still results in emissions too low to account for the missing source."

*3. It should also be clearly acknowledged that the modeled downwelling (or downward) irradiance (as described in the paper) are crude approximations and do not even include the effects of clouds for example. This is another important source of uncertainty.*

For the case studies during the cruises, we used measured radiation which includes the effect of clouds and the absorption by gases/aerosols in the atmosphere, as it was measured at the surface directly on the ship. For the global simulation, the effect of clouds and other absorption is indirectly considered. We use Era Interim data, where measured data is assimilated (Dee et al., 2011). Monthly climatologies of sunshine duration and daily incoming radiation, which include the effects of clouds and atmospheric absorption of aerosols and gases (Dee et al., 2011), are then used to fit mean monthly diurnal cycles of radiation. While the fitted diel cycle is always parabolic, the integral of the daily incoming radiation includes the effect of any attenuation in the atmosphere. The main uncertainty lies in the conversion to the UV light, which is counterbalanced by the photoproduction rate constant p that is optimized to this specific way of treating radiation in the model, across several regions and latitudes (tropical Indian Ocean, tropical Peruvian upwelling, Northern Atlantic ocean from the von Hobe et al. (2003) study). The fact that the model simulates concentrations of OCS in line with previous observations supports the validity of these simplifications.

*4. It should also be better mentioned that there are uncertainties in the a350 retrieved from satellite data and that uncertainty will be compounded when p is derived from a350. There is also uncertainty due to the fact that the GSM model used to retrieved the absorption by detritus+CDOM at 443 nm and not just CDOM. Finally, another uncertainty is associated with the use of a single spectral slope for CDOM to derive a350 from a443. All these combined uncertainties can amount to a large error in the estimates of OCS photoproduction alone, and I think there is a need to better acknowledge all sources of uncertainty and their implications on the conclusions, and there is a need for a more cautious language with regards to the conclusions of this work.*

We agree that pointing out these uncertainties is important and add a more detailed discussion on this in the new section "3.2.2. Uncertainties". However, assuming a443nm is only CDOM would lead to an overestimation and thus a conservative estimate in our case. As our comparison of the

resulting global simulation with measurements shows, our simulated concentrations agree very well with previous and independent observations. A large error through these combined uncertainties thus seems unlikely. Given that the direct emissions are a factor of ~5 lower than what would be needed to account for the missing source, and that the previous measurements support our simulated concentrations, these uncertainties do not interfere with the main conclusion.

*I also would like to take the opportunity to use the attached figures of COS photoproduction rates integrated over the mixed layer to highlight that the regions studied in your paper are generally not as photochemically active (generally lower photoproduction rates) as the region studied in Mihalopoulos et al. (1992). This is in relation to the comment posted by Dr. Belviso.*

Thank you for providing these plots of your model calculations which shows the photoproduction rates if a globally constant, i.e. not CDOM dependent, photoproduction rate constant (or quantum yield), is assumed. This is different to our approach, where the amount of CDOM has a double effect on the photoproduction rate and, thus, shows a different spatial pattern, with hot spots in regions of high CDOM, i.e. in the Peruvian upwelling where we carried out our second study. The chemical background for a CDOM dependent photoproduction rate constant is described in detail in von Hobe et al. (2003). Still, including these two regions together with the photoproduction rate constants from the Atlantic region into one global p-CDOM-relationship yielded simulated oceanic concentrations of OCS that are in line with observations across different ocean basins and latitudes (Tab. 6 in revised manuscript).